# Efficient Availability Attacks against Supervised and Contrastive Learning Simultaneously

**Yihan Wang, Yifan Zhu, Xiao-Shan Gao**[*]
Academy of Mathematics and Systems Science, Chinese Academy of Sciences
University of Chinese Academy of Sciences
{yihanwang, zhuyifan}@amss.ac.cn, xgao@mmrc.iss.ac.cn

## Abstract

Availability attacks provide a tool to prevent the unauthorized use of private data and commercial datasets by generating imperceptible noise and crafting unlearnable examples before release. Ideally, the obtained unlearnability can prevent algorithms from training usable models. When supervised learning (SL) algorithms have failed, a malicious data collector possibly resorts to contrastive learning (CL) algorithms to bypass the protection. Through evaluation, we have found that most existing methods are unable to achieve both supervised and contrastive unlearnability, which poses risks to data protection by availability attacks. Different from recent methods based on contrastive learning, we employ contrastive-like data augmentations in supervised learning frameworks to obtain attacks effective for both SL and CL. Our proposed AUE and AAP attacks achieve state-of-the-art worst-case unlearnability across SL and CL algorithms with less computation consumption, showcasing prospects in real-world applications. The code is available at `https://github.com/EhanW/AUE-AAP`.

## 1 Introduction

Availability attacks [2] add imperceptible perturbations to training data, making the subsequently trained model unavailable. The motivations behind this kind of data poisoning attack involve protecting private data and commercial datasets from unauthorized use [22]. For example, according to a report [19], a tech company illegally obtained over 3B facial images as the training set to develop a commercial facial recognition model. In this type of scenario, availability attacks provide tools to process user images before release, preserving legibility but impeding subsequent training. In particular, Huang et al. [22] reduces the accuracy of face recognition of 50 identities in WebFace [54] from 86% to 16%. In recent years, various availability attacks against supervised learning (SL) [11–13, 41, 56, 28] have been proposed.

Meanwhile, contrastive learning (CL) allows people to extract meaningful features from unlabeled data in a self-supervised way. After subsequent linear probing or fine-tuning, CL algorithms have achieved comparable accuracy or even surpassed the performance of SL [5, 7, 15, 6]. However, most attacks designed for poisoning SL are ineffective against CL, as shown in Figure 1. It sheds light on a potential issue of using availability attacks to protect data: a malicious data collector can traverse both supervised and contrastive algorithms to effectively leverage collected data. Hence, we introduce *worst-case unlearnability* (see Section 3.1) as the evaluation metric for availability attacks to emphasize the demand to deal with a trickier unauthorized data

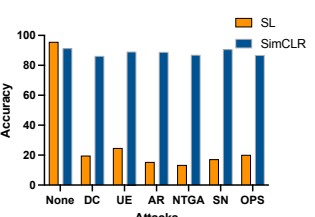

Figure 1: Attacks against SL and CL on CIFAR-10.

---

[*]Corresponding author. Kaiyuan International Mathematical Sciences Institute

38th Conference on Neural Information Processing Systems (NeurIPS 2024).

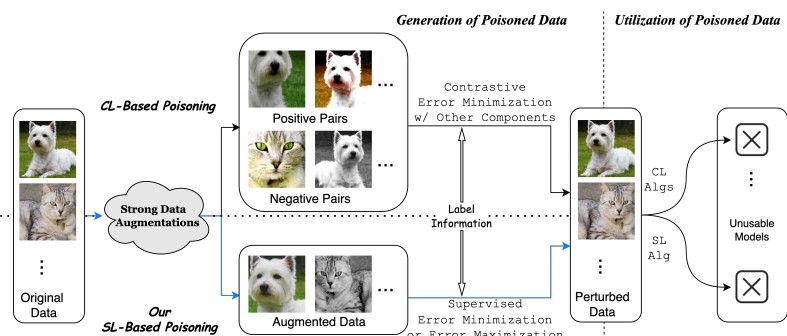

Figure 2: Illustration of our proposed method. Separated by a vertical dashed line, the left side shows the process of generating the poisoning attack, while the right side depicts the training process on the poisoned dataset. On the generation side, above the horizontal dashed line are the existing methods based on contrastive error minimization, while below the dashed line are our proposed methods based on supervised error minimization/maximization (the blue flow). Our attack leverages the stronger contrastive augmentations to obtain effectiveness against both supervised learning and contrastive learning algorithms. Label information is involved in both our method and CL-based methods.

collector. In recent years, contrastive error minimization attack is proposed to poison contrastive learning [16], and then label-dependent components are incorporated into it to simultaneously achieve supervised unlearnability besides contrastive unlearnability [36, 29]. However, compared to SL-based ones, these CL-based attacks lack efficiency in poisoning generation, potentially hindering availability attacks from protecting extensive data in the real world (see Section 5.3). Therefore, an effective as well as efficient availability attack against both SL and CL is imminent. Specifically, our motivation for this paper comes from two aspects: 1) *A fully functional availability attack needs to be effective against subsequent supervised and contrastive learning algorithms simultaneously.* 2) *Attacks based on supervised learning can be superior in efficiency compared to those based on contrastive learning.*

To design a non-CL-based attack that possesses both supervised and contrastive unlearnability, we start from an interesting observation that supervised training with contrastive data augmentations mimics contrastive training to some extent (see Section 4.1). As shown in Figure 2, this technique of enhancing data augmentations can be easily embodied in two basic supervised attack frameworks, i.e., error-minimization, and error-maximization, resulting in our proposed AUE and AAP attacks (see Sections 4.2 and 4.3). Enhanced augmentations allow us to craft perturbations for a contrastive-like reference model. These perturbations im-

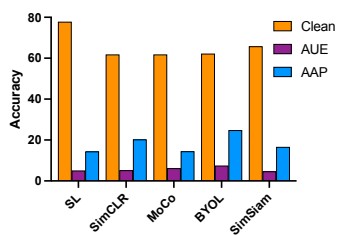

Figure 3: Attack performance of our methods on ImageNet-100.

plicitly adapt to the contrastive training process and then learn deceptive patterns that fool contrastive learning. The supervised unlearnability is still preserved since the generation process is based on supervised optimization.

On experimental side, we evaluate the standard supervised learning algorithm and four representative contrastive learning algorithms, SimCLR [5], MoCo [7], BYOL [15] and SimSiam [6]. Our proposed AUE and AAP attacks achieve the state-of-the-art worst-case unlearnability on CIFAR-10/100 and Tiny/Mini-ImageNet datasets (see Section 5.2). Specifically, our method exhibits excellent performance on the ImageNet-100 as presented in Figure 3, showcasing its prospects in real-world applications. Meanwhile, unlike methods that add additional components to the contrastive error-minimization framework, we modify the data augmentation in the simpler supervised attack frameworks, following a minimalist approach to algorithm design. Benefiting from this, our methods are more efficient, while delivering better performance. We summarize our contributions as follows:

- We evaluate existing availability attacks and point out the potential security risks of using them to protect data when facing data abusers who will traverse both supervised and contrastive learning algorithms.

- We start from supervised poisoning approaches and enhance data augmentations to attain attacks against both supervised and contrastive learning.

- Our attacks achieve state-of-the-art worst-case unlearnability with less computation consumption and are more adept at handling high-resolution datasets.

## 2 Background

We will introduce some notions of contrastive learning and availability attacks. Besides, we provide more preliminaries on contrastive learning in Appendix B.

### 2.1 Contrastive Learning

Contrastive learning trains feature encoders in a self-supervised way. It first transforms an image into two views, i.e., a positive pair, using augmentations sampled from a strong augmentation distribution $\mu$. Two views augmented from different images constitute a negative pair. Extracted features are trained to be aligned between positive pairs but distinct between negative pairs. It does not require label information until downstream tasks such as linear probing or fine-tuning. Wang and Isola [50] introduced two key properties for contrastive learning, *alignment* and *uniformity*. The former measures the similarity of features from positive pairs and the latter reflects the uniformity of feature distribution on the hypersphere. Let $g$ be a normalized encoder. The *alignment loss* and *uniformity loss* on a dataset $\mathcal{D}_c$ are defined as the following:

$$\mathcal{A}(\mathcal{D}_c) = \mathop{\mathbb{E}}_{\substack{\boldsymbol{x} \sim \mathcal{D}_c \\ \pi, \tau \sim \mu}} \left[ ||g(\pi(\boldsymbol{x})) - g(\tau(\boldsymbol{x}))||_2^2 \right]; \quad \mathcal{U}(\mathcal{D}_c) = \log \mathop{\mathbb{E}}_{\substack{\boldsymbol{x}, \boldsymbol{z} \sim \mathcal{D}_c \\ \pi, \tau \sim \mu}} \left[ e^{-2||g(\pi(\boldsymbol{x})) - g(\tau(\boldsymbol{z}))||_2^2} \right].$$

Let $\mathcal{D}'_c$ be a poisoned version of a clean dataset $\mathcal{D}_c$. The *alignment gap* and *uniformity gap* between clean and poisoned datasets are defined as follows:

$$\mathcal{AG} = \mathcal{A}(\mathcal{D}_c) - \mathcal{A}(\mathcal{D}'_c), \quad \mathcal{UG} = \mathcal{U}(\mathcal{D}_c) - \mathcal{U}(\mathcal{D}'_c). \tag{1}$$

Intuitively, these gaps characterize the difference between clean features and poisoned features. We will check the relationship between these gaps and contrastive unlearnability in Section 3.2.

### 2.2 Basic Availability Attacks

The essence of availability attacks is to prevent a trained model from well generalizing to clean data. We will revisit two representative approaches to poisoning supervised learning.

**Error minimization.** Unlearnable example attack (UE) generates poisoning by alternately optimizing the reference model and perturbations [22]:

$$\min_{\delta} \min_{f} \mathop{\mathbb{E}}_{\mathcal{D}_c} \left[ \mathcal{L}_{\text{SL}}(\boldsymbol{x} + \delta(\boldsymbol{x}, y), y; f) \right], \tag{2}$$

where $f$ is a classifier, $\mathcal{L}_{\text{SL}}(\cdot, \cdot; \cdot)$ is the supervised loss, $\mathcal{D}_c$ is a dataset to be processed and $\delta$ is a poisoning map.

**Error maximization.** Adversarial poisoning (AP) optimizes perturbations through a pre-trained classifier to equip them with non-robust but useful features from a different label [13]:

$$\min_{\delta} \mathop{\mathbb{E}}_{\mathcal{D}_c} \left[ \mathcal{L}_{\text{SL}}(\boldsymbol{x} + \delta(\boldsymbol{x}, y), y + K; f^*) \right], \tag{3}$$

$$\text{s.t.} \quad f^* \in \arg\min_{f} \mathop{\mathbb{E}}_{\mathcal{D}_c} \left[ \mathcal{L}_{\text{SL}}(\boldsymbol{x}, y; f) \right],$$

where $K$ is the label shift. Self-ensemble protection (SEP) generates adversarial poisons using several checkpoints to improve attack performance [4].

**Contrastive error minimization.** Contrastive poisoning (CP) extends the error minimization framework to contrastive error minimization to poison contrastive learning [16]:

$$\min_{\delta} \min_{g} \mathop{\mathbb{E}}_{\mathcal{D}_c} \left[ \mathcal{L}_{\text{CL}}(\boldsymbol{x} + \delta(\boldsymbol{x}, y); g) \right], \tag{4}$$

where $g$ is an encoder and $\mathcal{L}_{\text{CL}}(\cdot; \cdot)$ denotes the contrastive loss for simplicity. Later, the transferable unlearnable example attack (TUE) introduces a regularization term called class-wise separability discriminant to equip CP noises with supervised unlearnability [36]. Then, transferable poisoning (TP) combines contrastive error minimization with supervised adversarial poisoning to obtain both supervised and contrastive unlearnability [29]. It is worth mentioning that both TUE and TP leverage label information in their proposed schemes, while CP requires no label information but lacks stable effect on supervised learning.

# 3 Threat Model

In our threat model, an unauthorized data collector assembles labeled data into a dataset. The access to label information is reasonable since the collector can crawl individual images from certain accounts or steal (and annotate) a commercial dataset. A data publisher is supposed to process data before release using an availability attack such that processed data is resilient to subsequent supervised learning algorithms as well as contrastive learning algorithms adopted by the data collector. We will define worst-case unlearnability and discuss the contrastive unlearnability of existing attacks.

## 3.1 Worst-Case Unlearnability

Suppose an unprocessed dataset $\mathcal{D}_c$ is *i.i.d* sampled from a data distribution $\mathcal{D}$. An availability attack $\delta$ maps a data-label pair $(\boldsymbol{x}, y) \in \mathcal{D}_c$ to a noise $\delta(\boldsymbol{x}, y)$ within an $L_p$-norm ball $\mathcal{B}_p(\epsilon)$. In this paper, we set $p = \infty$ and $\epsilon = 8/255$. It results in a protected dataset $\mathcal{D}'_c = \{(\boldsymbol{x} + \delta(\boldsymbol{x}, y), y) | (\boldsymbol{x}, y) \in \mathcal{D}_c\}$ to which a data collector has only access. For potential algorithms, we refer $f$ to a supervised learning classifier and $g$ to a contrastive learning encoder beyond which is a linear probing head $h$. The goal of the data publisher is to find a poisoning map $\delta$ that significantly degrades the generalization performance of both $f_\delta$ and $h_\delta \circ g_\delta$ which are trained on $\mathcal{D}'_c$. We define the *worst-case unlearnability across supervised and contrastive learning algorithms* of the following form:

$$\min_\delta \max(\mathbb{E}_{\mathcal{D}} \left[ \mathbf{1}(f_\delta(\boldsymbol{x}) = y) \right], \mathbb{E}_{\mathcal{D}} \left[ \mathbf{1}(h_\delta \circ g_\delta(\boldsymbol{x}) = y) \right]) \qquad (5)$$

$$\text{s.t.} \quad f_\delta \in \arg\min_f \mathbb{E}_{\mathcal{D}_c} \left[ \mathcal{L}_{\text{SL}}(\boldsymbol{x} + \delta(\boldsymbol{x}, y), y; f) \right],$$

$$g_\delta \in \arg\min_g \mathbb{E}_{\mathcal{D}_c} \left[ \mathcal{L}_{\text{CL}}(\boldsymbol{x} + \delta(\boldsymbol{x}, y); g) \right],$$

$$h_\delta \in \arg\min_h \mathbb{E}_{\mathcal{D}_c} \left[ \mathcal{L}_{\text{SL}}(\boldsymbol{x} + \delta(\boldsymbol{x}, y), y; h \circ g_\delta) \right].$$

It is a fair metric that accurately depicts scenarios facing more cunning data abusers in the real world. In contrast, other metrics, such as average-case unlearnability, can be heavily influenced by the attack's strong preference for a certain algorithm. Our threat model differs from the setting adopted by He et al. [16] in which the linear probing stage relies on the unprocessed clean data as downstream tasks; see more discussion in Appendix D.10.

## 3.2 Existing Attacks against Contrastive Learning

In Table 1, we evaluate the attack performance of existing poisoning approaches against the SimCLR algorithm on CIFAR-10 and ResNet-18. To better understand contrastive unlearnability, we also check alignment and uniformity gaps between clean and poisoned data defined in Equation (1). In non-CL-based poisoning attacks, except for AP and SEP, all other methods fail to deceive the contrastive learning algorithm. The alignment and uniformity gaps of AP and SEP attacks are prominently larger than those of others. CL-based attacks including CP, TUE, and TP are effective against contrastive learning and possess huge alignment and uniformity gaps.

The Pearson correlation coefficient (PCC) between the alignment gap and SimCLR accuracy is $-0.82$, and the PCC between the uniformity gap and Sim-CLR accuracy is $-0.88$. It reveals that contrastive unlearnability is highly related to huge alignment and uniformity gaps. When the encoder is fixed after poisoned contrastive training, a linear layer learns to classify poisoned features, i.e., features of poisoned (training) data. Note that evaluation is to classify clean features, i.e., features of clean (test) data.

Table 1: Alignment gap, uniformity gap, and test accuracy(%) of poisoned SimCLR [5] models. Attacks are grouped according to whether they are based on contrastive error minimization. **Bold** fonts emphasize prominent contrastive unlearnability values.

| Attacks | $\mathcal{AG}$ | $\mathcal{UG}$ | Test Acc. |
|---|---|---|---|
| DC [11] | 0.12 | 0.07 | 86.1 |
| UE [22] | 0.05 | 0.03 | 89.0 |
| AR [41] | 0.07 | 0.09 | 88.8 |
| NTGA [58] | 0.12 | 0.12 | 86.9 |
| SN [56] | 0.08 | 0.00 | 90.6 |
| OPS [52] | 0.04 | 0.01 | 86.7 |
| GUE [28] | 0.07 | 0.03 | 88.8 |
| REM [14] | 0.12 | 0.04 | 88.6 |
| EntF [51] | 0.01 | -0.04 | 87.5 |
| HYPO [47] | 0.11 | 0.13 | 86.9 |
| AP [13] | **0.18** | **0.44** | **48.4** |
| SEP [4] | **0.24** | **0.25** | **37.3** |
| CP [16] | **0.55** | **0.87** | **38.7** |
| TUE [36] | **0.30** | **0.76** | **48.1** |
| TP [29] | **0.52** | **0.82** | **31.4** |

Prominent gaps indicate a significant difference between clean and poisoned feature distributions. Thus, no matter how well the classifier performs on poisoned features, it hardly generalizes to clean features and the attack is successful. In contrast, small gaps likely imply that clean features are similar to poisoned features. When gaps are small, the test accuracy is high and the attack fails.

# 4 Method

Since contrastive loss is related to alignment and uniformity [50], the contrastive error minimization (CP) attack optimizes the loss directly and obtains contrastive unlearnability. Beyond this, TUE and TP incorporate additional label-dependent components to obtain supervised unlearnability. However, optimizing contrastive loss is very time and memory-consuming, impeding their applications in real-world scenarios. Different from them, we start from more efficient supervised poisoning frameworks instead to achieve both supervised unlearnability and contrastive unlearnability simultaneously. The key point to get there is data augmentation. In the rest of this section, we first illustrate how contrastive learning data augmentations help mimic contrastive learning with supervised models through empirical observations and intuition from a toy example. In other words, enhancing data augmentation helps supervised learning implicitly optimize the contrastive loss. Then we combine this very effective technique with supervised error minimization and maximization frameworks and propose *augmented unlearnable examples (AUE) attacks* and *augmented adversarial poisoning (AAP) attacks*.

## 4.1 Mimic Contrastive Learning with Supervised Models

Contrastive learning employs strong data augmentations including *resized crop, color jitter, horizontal flip, and grayscale* [53, 18], while supervised learning adopts mild data augmentations such as horizontal flip and crop. In Appendix C.2, Code 1 shows detailed implementations for these two different settings. Naturally, contrastive error minimization uses stronger data augmentation compared to supervised error minimization. However, what if we use strong contrastive augmentations when optimizing supervised losses? On CIFAR-10, we train a supervised ResNet-18 classifier using contrastive augmentations. At each checkpoint, we log the supervised

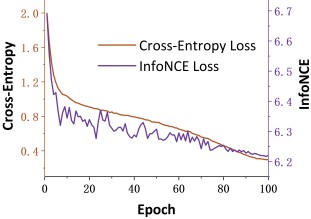

Figure 4: InfoNCE loss decreases with CE loss.

cross-entropy (CE) loss and the contrastive InfoNCE loss [33] on the training set. In Figure 4, when the optimization object CE loss goes down, the InfoNCE loss decreases as well. It indicates that training a supervised model with contrastive augmentations implicitly optimizes the contrastive loss. In other words, supervised error minimization mimics contrastive error minimization to some extent. Therefore, incorporating stronger data augmentation potentially enables availability attacks based on supervised error minimization or maximization to deceive contrastive learning.

To provide more intuition about this idea, we give a toy example and have a closer look at the relationship between supervised loss and contrastive loss. For a supervised model $f = h \circ g$, assume $g$ is a normalized feature extractor, $h$ is a square full-rank linear classifier, $\mathcal{D}$ is a balanced distribution, $\mathcal{L}_{SL}$ is MSE loss, training error $\mathcal{E}_{SL} = \mathbb{E}\left[\mathcal{L}_{SL}\right]$, and $\mathcal{L}_{CL}$ contains only one negative example. In this toy example, if $\mathcal{L}_{CL}$ and $\mathcal{L}_{SL}$ employ the same data augmentation and $f$ is well-trained, it holds with high probability that $\mathcal{L}_{CL} < l(\mathcal{E}_{SL})$, where $l(\cdot)$ is an increasing function. In other words, the upper bound of contrastive loss decreases as the supervised loss decreases. We have a more detailed and formal discussion on this toy example in Appendix E.

Based on these interesting observations, instead of adding components to contrastive error minimization to achieve supervised unlearnability, we opt for deriving stronger contrastive unlearnability from supervised error minimization and maximization.

## 4.2 Augmented Unlearnable Examples (AUE)

Recall unlearnable examples (UE) are generated by supervised error minimization which alternately updates a reference model and noises in Equation (2). Now we employ contrastive-like strong data augmentation distribution $\mu$ and add perturbations in a differentiable way, i.e., $\pi(\boldsymbol{x} + \delta(\boldsymbol{x}, y)), \pi \sim \mu$. As discussed in the previous section, minimizing the augmented supervised loss $\mathcal{L}_{SL}(\pi(\boldsymbol{x} + \delta(\boldsymbol{x}, y)), y; f)$

implicitly minimizes the contrastive loss $\mathcal{L}_{\text{CL}}(\boldsymbol{x} + \delta(\boldsymbol{x}, y); g)$ which appears in contrastive error minimization, i.e., Equation (4). In other words, supervised error-minimizing noises with enhanced data augmentation can partially replace the functionality of contrastive error-minimizing noises to deceive contrastive learning.

We can control the intensity of contrastive augmentations in Code 1 via a strength parameter $s \in [0, 1]$. We increase the augmentation strength and generate the poisoning attack using Algorithm 1. Implementation details are shown in Appendix C.3. In Table 2, while UE attacks do not work for SimCLR on CIFAR-10 and CIFAR-100, our AUE attacks successfully reduce the SimCLR accuracy by $38.9\%$ and $50.3\%$. Enhanced data augmentation indeed makes supervised error-minimizing noises effective for contrastive learning. In Figure 5a, AUE noises largely reduce the contrastive loss

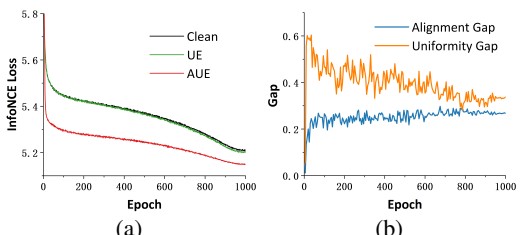

Figure 5: (a) Contrastive losses during SimCLR training under UE and AUE attacks. (b) Alignment and uniformity gaps during the SimCLR training on CIFAR-10 poisoned by our AUE attack.

during SimCLR training compared to UE noises. In Figure 5b, we investigate the alignment and uniformity gaps and discuss more about the poisoned training process in Appendix D.2. The final gaps of AUE are $\mathcal{AG} = 0.27, \mathcal{UG} = 0.34$ while those of UE are $\mathcal{AG} = 0.05, \mathcal{UG} = 0.03$.

---

**Algorithm 1** Augmented Unlearnable Examples (AUE)

**Require:** Augmentation strength $s$ and a corresponding augmentation distribution $\mu_s$. A labeled training set $\mathcal{D}_c = \{(\boldsymbol{x}_i, y_i)\}_{i=1}^r$. An initialized classifier $f_\theta$. Total epochs $T$, model update iterations $T_\theta$, poisons update iterations $T_\delta$, and perturbation steps $T_p$. Learning rate $\alpha_\theta, \alpha_\delta$.
**Ensure:** Perturbations $\{\boldsymbol{\delta}_i\}_{i=1}^r$
$\quad \boldsymbol{\delta}_i \leftarrow 0, i = 1, 2, \cdots, r$ $\hfill \triangleright$ Initialize perturbations
$\quad$ **for** $t = 1, \cdots, T$ **do**
$\quad\quad$ **for** $t_\theta = 1, \cdots, T_\theta$ **do** $\hfill \triangleright$ Update the reference model
$\quad\quad\quad$ Sample a data batch $\{(\boldsymbol{x}_{l_j}, y_{l_j})\}_{j=1}^m$ and an augmentation batch $\{\pi_{l_j} \sim \mu_s\}_{j=1}^m$
$\quad\quad\quad$ $\theta \leftarrow \theta - \frac{\alpha_\theta}{m} \cdot \sum_{j=1}^m \nabla_\theta \mathcal{L}_{\text{SL}}(\pi_{l_j}(\boldsymbol{x}_{l_j} + \boldsymbol{\delta}_{l_j}), y_{l_j}; f_\theta)$
$\quad\quad$ **for** $t_\delta = 1, \cdots, T_\delta$ **do** $\hfill \triangleright$ Update perturbations
$\quad\quad\quad$ Sample a data batch $\{(\boldsymbol{x}_{l_j}, y_{l_j})\}_{j=1}^m$
$\quad\quad\quad$ **for** $t_p = 1, \cdots, T_p$ **do**
$\quad\quad\quad\quad$ Sample an augmentation batch $\{\pi_{l_j} \sim \mu_s\}_{j=1}^m$
$\quad\quad\quad\quad$ $\boldsymbol{\delta}_{l_j} \leftarrow \text{Clip}_\epsilon \left( \boldsymbol{\delta}_{l_j} - \alpha_\delta \cdot \text{sign}(\nabla_{\boldsymbol{\delta}_{l_j}} \mathcal{L}_{\text{SL}}(\pi_{l_j}(\boldsymbol{x}_{l_j} + \boldsymbol{\delta}_{l_j}), y_{l_j}; f_\theta)) \right), j = 1, 2, \cdots, m$

---

### 4.3 Augmented Adversarial Poisoning (AAP)

Adversarial poisoning (AP) attacks in Equation (3) first train a supervised reference model, then generate its adversarial examples. When replacing mild supervised augmentations with stronger contrastive ones, the training process of the reference model, i.e., minimizing $\mathcal{L}_{\text{SL}}(\pi(\boldsymbol{x}), y; f), \pi \sim \mu$ concerning $f$ mimics

Table 2: Accuracy drop(%) of SimCLR caused by basic attacks and our methods.

| Datasets | Clean | UE | AUE | AP | AAP |
|---|---|---|---|---|---|
| CIFAR-10 | 91.3 | -2.3 | -38.9 | -42.9 | -52.2 |
| CIFAR-100 | 63.9 | -3.9 | -50.3 | -38.3 | -43.8 |

updating its encoder with contrastive learning. The final reference model $f^*$ has a contrastive-like encoder. Then, generating perturbations via minimizing $\mathcal{L}_{\text{SL}}(\pi(\boldsymbol{x} + \delta(\boldsymbol{x}, y)), y + K; f^*)$ with respect to $\delta$ is to deceive the contrastive-like model. Consequently, the resulting poisoning attack learns more about confounding contrastive learning algorithms.

According to Algorithm 2, we increase the augmentation strength $s$ in both reference model pre-training and noise update where the label translation $K = 1$. Implementation details are shown in Appendix C.3. In Table 2, the AAP attack further enlarges the SimCLR accuracy drop of AP by $9.3\%$ on CIFAR-10 and $5.5\%$ on CIFAR-100. Enhanced data augmentations indeed improve the contrastive unlearnability of supervised error-maximizing noises.

---
**Algorithm 2** Augmented Adversarial Poisoning (AAP)
---
**Require:** Similar to the setting in Algorithm 1.
**Ensure:** Perturbations $\{\boldsymbol{\delta}_i\}_{i=1}^r$
    $\boldsymbol{\delta}_i \leftarrow 0, i = 1, 2, \cdots, r$                                          ▷ Initialize perturbations
    **for** $t = 1, \cdots, T$ **do**                                   ▷ Update the reference model
        **for** $t_\theta = 1, \cdots, T_\theta$ **do**
            Sample a data batch $\{(\boldsymbol{x}_{l_j}, y_{l_j})\}_{j=1}^m$ and an augmentation batch $\{\pi_{l_j} \sim \mu_s\}_{j=1}^m$
            $\theta \leftarrow \theta - \frac{\alpha_\theta}{m} \cdot \sum_{j=1}^m \nabla_\theta \mathcal{L}_{\mathrm{SL}}(\pi_{l_j}(\boldsymbol{x}_{l_j}), y_{l_j}; f_\theta)$
    **for** $i = 1, \cdots, r$ **do**                                     ▷ Update adversarial examples
        **for** $t_p = 1, \cdots, T_p$ **do**
            Sample $\pi_i \sim \mu_s$
            $\boldsymbol{\delta}_i \leftarrow \mathrm{Clip}_\epsilon\big(\boldsymbol{\delta}_i - \alpha_\delta \cdot \mathrm{sign}(\nabla_{\boldsymbol{\delta}_i} \mathcal{L}_{\mathrm{SL}}(\pi_i(\boldsymbol{x}_i + \boldsymbol{\delta}_i), y_i + 1; f_\theta)))$
---

# 5 Experiments

We will evaluate the worst-case unlearnability of our proposed AUE and AAP attacks on multiple datasets and compare the poisoning generation efficiency with other methods. Besides, we will check the efficacy of our method against more evaluation algorithms and the transferability across network architectures. Then we will perform an ablation study of decoupling argumentation components in our method.

## 5.1 Setup

We conduct experiments on CIFAR-10/100 [26], Tiny-ImageNet [27], modified Mini-ImageNet [49], and ImageNet-100 [38]. ResNet-18 [17] is used for poison generation and evaluation if not otherwise stated. Our threat model considers the worst-case unlearnability across supervised and contrastive (self-supervised) algorithms including standard SL, SimCLR, MoCo, BYOL, and SimSiam. We implement linear probing on the encoder to evaluate contrastive unlearnability.

We adopt AP, SEP-FA-VR, CP, TUE, and TP as baseline methods for the worst-case unlearnability. Since the generation of untargeted adversarial poisoning is unstable [13], AP and AAP attacks are targeted if not otherwise stated (see more discussion in Appendix D.5). In particular, only CIFAR-10 results in Table 3 report untargeted AP and AAP. For CP and TUE attacks, we report the best results across the CL algorithms they depend on (see additional results in Appendix D.6). Detailed settings for attack implementation and evaluation are shown in Appendix C.

Table 3: Attack Performance (%) on CIFAR-10 and CIFAR-100. The lower the value, the better the unlearnability.

| Attacks | CIFAR-10 | | | | | | CIFAR-100 | | | | | |
|---|---|---|---|---|---|---|---|---|---|---|---|---|
| | SL | SimCLR | MoCo | BYOL | SimSiam | Worst | SL | SimCLR | MoCo | BYOL | SimSiam | Worst |
| None | 95.5 | 91.3 | 91.5 | 92.3 | 90.7 | 95.5 | 77.4 | 63.9 | 67.9 | 63.7 | 64.4 | 77.4 |
| AP | 9.6 | 41.5 | 31.5 | 44.0 | 42.8 | 44.0 | 3.2 | 25.6 | 26.6 | 26.1 | 28.8 | 28.8 |
| SEP | 2.3 | 37.3 | 35.8 | 42.8 | 36.7 | 42.8 | 2.4 | 25.2 | 25.9 | 26.6 | 28.4 | 28.4 |
| CP | 11.0 | 39.3 | 32.7 | 41.8 | 37.9 | 41.8 | 74.4 | 15.2 | 13.4 | 16.4 | 14.1 | 74.4 |
| TUE | 10.1 | 57.2 | 51.6 | 60.1 | 58.5 | 60.1 | 1.0 | 19.9 | 19.6 | 22.3 | 18.6 | 22.3 |
| TP | 14.8 | 31.4 | 54.1 | 61.8 | 30.7 | 61.8 | 7.5 | 6.7 | 21.9 | 27.0 | 4.1 | 27.0 |
| AAP | 29.7 | 32.3 | 23.2 | 35.5 | 34.1 | **35.5** | 7.3 | 20.1 | 18.6 | 21.1 | 21.3 | 21.3 |
| AUE | 18.9 | 52.4 | 57.0 | 58.2 | 34.5 | 58.6 | 6.9 | 13.6 | 19.0 | 19.2 | 11.9 | **19.2** |

## 5.2 Attack Performance

**Worst-case unlearnability.** In Table 3, our AAP attack achieves the best worst-case unlearnability on CIFAR-10 and both AAP and AUE attacks outperform other methods on CIFAR-100. Particularly, AAP improves the performance by 8.5%/7.5% on CIFAR-10/100 and AUE becomes better than AAP on CIFAR-100. In other methods, CP loses supervised unlearnability on CIFAR-100 and TUE is better than TP on both datasets. Furthermore, we evaluate attacks on higher-resolution datasets including Tiny-ImageNet (64x64) and Mini-ImageNet (84x84) in Table 4. On both datasets, AUE outperforms other methods in terms of the worst-case unlearnability. Besides, AUE also achieves the best supervised unlearnability. For AAP, its worst-case unlearnability is better than AP but not as

good as TUE. Moreover, compared to AP, AAP suffers a trade-off between supervised unlearnability and contrastive unlearnability.

**Comparison between AUE and AAP.** On simpler datasets (low resolution, few classes), AAP has an advantage over AUE. However, on more complex datasets (high resolution, many classes), AUE outperforms AAP. One possible reason for this could be that optimizing AAP is more challenging. Firstly, generating adversarial poisoning inherently depends on a well-performing reference classifier. For instance, Fowl et al. 13 uses a pre-trained ImageNet model, whereas in this paper we train classifiers from scratch. Additionally, stronger data augmentation used in the reference model training can affect its accuracy, which in turn impacts the quality of the generated adversarial perturbations. Enhancing the performance of AAP is an interesting direction for future work.

**Algorithm transferability.** CL-based methods face the issue of transferability from the generation CL algorithm and the evaluation CL algorithm. For example, TP is generated using the SimCLR, which performs well against SimCLR evaluation, but its performance sharply declines when tested with BYOL. The same phenomenon also occurs with TUE and CP and their worst-case unlearnability is highly dependent on the appropriate generation algorithm, which you can check in Appendix D.6. In contrast, our SL-based attacks get rid of this issue because their poisoning generation involves no CL algorithms.

Table 4: Attack Performance (%) on Mini-ImageNet and Tiny-ImageNet. The lower the value, the better the unlearnability.

| Attacks | MINI-IMAGENET | | | | | | TINY-IMAGENET | | | | | |
| --- | --- | --- | --- | --- | --- | --- | --- | --- | --- | --- | --- | --- |
| | SL | SimCLR | MoCo | BYOL | SimSiam | Worst | SL | SimCLR | MoCo | BYOL | SimSiam | Worst |
| None | 66.2 | 55.3 | 57.6 | 48.7 | 54.5 | 66.2 | 53.5 | 39.6 | 43.3 | 33.9 | 42.4 | 53.5 |
| AP | 11.5 | 48.9 | 50.1 | 44.0 | 48.5 | 50.1 | 11.3 | 32.8 | 34.7 | 27.2 | 34.5 | 34.7 |
| TUE | 20.7 | 20.6 | 21.1 | 20.8 | 21.2 | 21.2 | 8.5 | 13.3 | 15.9 | 13.4 | 14.1 | 15.9 |
| AUE | 8.7 | 15.0 | 20.4 | 14.5 | 18.2 | **20.4** | 7.1 | 10.8 | 11.7 | 9.6 | 11.6 | **11.7** |
| AAP | 24.0 | 43.8 | 41.9 | 40.2 | 41.8 | 43.8 | 18.7 | 28.4 | 27.6 | 25.2 | 28.2 | 28.4 |

## 5.3 Efficiency of Poisoning Generation

In real-world scenarios, availability attacks need to generate perturbations for accumulating data as quickly as possible. For expanding datasets, like continually updated social media user data, the poisoning used for data protection also needs to be updated periodically. Since contrastive learning involves larger batches (e.g., 512) and a longer training process (e.g., 1000 epochs), these contrastive error minimization-based attacks require more time and memory consumption to generate perturbations.

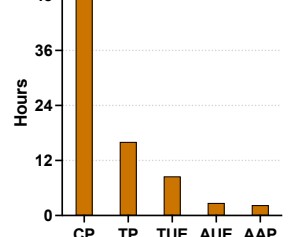

Figure 6: Time consumption of poisoning generation.

In Figure 6, we report the time cost of poisoning CIFAR-10/100 using the same device. Baseline methods adopt their default configurations. Our supervised learning-based approaches are 3x, 6x, and 17x faster than TUE, CP, and TP. Additionally, our methods admit smaller batches and simpler cross-entropy loss which require less memory, allowing for the generation of availability attacks on larger datasets with fewer devices. Refer to Appendix D.1 for more results about the efficiency of our methods. Overall, our method is more promising than CL-based methods due to the time and memory efficiency in real-world applications.

Table 5: Attack performance (%) against SimCLR $k$-NN, SupCL and FixMatch.

Table 6: Architecture transferability on CIFAR-10. Evaluation includes SL and SimCLR.

| Attacks | CIFAR-10 | | | CIFAR-100 | | |
| --- | --- | --- | --- | --- | --- | --- |
| | $k$-NN | SupCL | FixMatch | $k$-NN | SupCL | FixMatch |
| Clean | 88.9 | 94.6 | 95.7 | 55.2 | 72.5 | 77.0 |
| AUE | 54.4 | 31.5 | 30.0 | **13.3** | **15.6** | **12.0** |
| AAP | **42.6** | **24.7** | **18.7** | 21.7 | 17.9 | 25.5 |

| Alg. | Attacks | ResNet-50 | VGG | DenseNet | MobileNet | ViT |
| --- | --- | --- | --- | --- | --- | --- |
| SL | AUE | 16.4 | 23.2 | 19.5 | 17.2 | 33.4 |
| | AAP | 8.9 | 10.7 | 10.4 | 12.1 | 33.0 |
| CL | AUE | 53.4 | 48.2 | 50.5 | 41.4 | 45.1 |
| | AAP | 41.5 | 41.7 | 35.3 | 29.8 | 40.2 |

## 5.4 More Evaluation Algorithms

Besides linear probing, we also apply the k-nearest neighbors (k-NN) algorithm to the feature space to evaluate the contrastive unlearnability. In Table 5, both AUE and AAP prominently reduce the

k-NN accuracy. It indicates that the features of poisoned training inputs largely differ from those of clean test inputs. Non-robust features in imperceptible perturbations heavily affect the encoder's behavior and hinder its generalization ability.

In addition to supervised learning and contrastive learning algorithms, we consider two more CL-like algorithms including supervised contrastive learning, i.e., SupCL [24] and a semi-supervised learning algorithm FixMatch [45]. FixMatch uses WideResNet [59] and detailed settings are in Appendix C.5. Table 5 demonstrates that our attacks are still effective against SupCL and FixMatch. It indicates that our methods can handle more variants derived from supervised learning and contrastive learning algorithms.

## 5.5 Transferability across Networks

Since the data protector is unaware of networks used in future training, availability attacks should be effective for different architectures. We generate AUE and AAP using ResNet-18 and test them on ResNet-50, VGG-19 [44], DenseNet-121 [21], MobileNet v2 [20, 40], and ViT [10]. In Table 6, both supervised and contrastive unlearnability of AUE and AAP can transfer across these architectures.

## 5.6 Ablation Study of Decoupling Augmentations

In settings of AUE and AAP, we control the strength of ResizedCrop, ColorJitter, and Grayscale through a single strength hyperparameter $s$ for the poison generation, as shown in Code 1. In Table 7, we decouple the strength hyperparameters for these three random transforms and evaluate the resulting attacks against SimCLR. Different factors show different influences on the contrastive unlearnability for AUE and AAP. For example, enhancing ResizedCrop strength alone is less effective than enhancing Grayscale alone in AUE generation. However, adjusting three factors together generally outperforms other options in conclusion.

Table 7: SimCLR accuracy(%) of attacks generated with decoupled strength parameters on CIFAR-10. For example, 0-0-$s$ means that ResizedCrop strength is 0, ColorJitter strength is 0, and Grayscale strength is $s$.

| Attacks | 0-0-0 | 0-0-$s$ | 0-$s$-0 | $s$-0-0 | 0-$s$-$s$ | $s$-0-$s$ | $s$-$s$-0 | $s$-$s$-$s$ |
|---------|-------|---------|---------|---------|-----------|-----------|-----------|-------------|
| AUE | 83.5 | 58.7 | 79.4 | 88.7 | 60.8 | 56.2 | 87.7 | **52.4** |
| AAP | 52.3 | 52.0 | 52.9 | 44.9 | 51.4 | 42.2 | 44.8 | **39.1** |

## 6 Related Works

When generating availability attacks, the gradient of perturbations is often computed through data augmentations. In literature, SL-based attacks generally use mild supervised data augmentation, i.e., RandomCrop and RandomHorizontalFlip [13]. The expectation over transformation (EOT) technique [1] adopted by Fu et al. [14] first samples several such mild augmentations and then computes the average gradient over them. Note that our proposed method is not a variant of EOT. CL-based attacks use contrastive augmentations [16, 36, 29]. To our knowledge, we are the first to use contrastive-like strong data augmentations in SL-based poisoning frameworks. Besides, we provide additional related works on availability attacks in Appendix A.

## 7 Conclusion

Since contrastive learning algorithms bring new challenges to protect data using availability attacks, we explore effective attacks against both supervised and contrastive learning. We introduce a very effective modification of data augmentation in supervised poisoning frameworks and propose attacks achieving superior performance and efficiency compared to existing methods, offering more potential in real-world applications. Considering availability attacks still face obstacles such as adversarial training mitigation and poisoning ratio sensitivity, addressing these challenges while maintaining both supervised and contrastive unlearnability will be an important direction for our future research.

## Acknowledgments

This work is also supported by NKRDP grant No.2018YFA0704705, grant GJ0090202, NSFC grant No.12288201. The authors thank anonymous referees for their valuable comments.

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

# A  Additional Related Works

Availability attacks for supervised learning include error-minimizing noises [22], adversarial example poisoning [13, 4], neural tangent generalization attack [58], generative poisoning attack [11], autoregression perturbation [41], one-pixel perturbation [52], convolution-based attack [39], synthetic perturbation [56], and game-theoretic unlearnable examples [28]. Yu et al. [56] illustrated linearly separable perturbations work as shortcuts for supervised learning. Robust error-minimizing noises [14], entangled features strategy [51], and hypocritical perturbation [47] were designed to deceive adversarial training. Contrastive poisoning [16] aimed at poisoning contrastive learning. Transferable unlearnable examples [36] and transferable poisoning [29] improved the supervised unlearnability of contrastive poisoning. Zhang et al. [60] proposed to generate label-agnostic noises with cluster-wise perturbations. Chen et al. [3] introduced CLIP-guided unlearnable perturbation generators that can transfer across different datasets.

On the defense side, adversarial training can largely mitigate the unlearnability [46]. Liu et al. [30], Qin et al. [35], Zhu et al. [61] leverages crafted data augmentations as defense. Sandoval-Segura et al. [42] suggests that the orthogonal projection technique is effective against class-wise attacks. Diffusion models have been proposed to purify unlearnable perturbations [23, 9]. Yu et al. [57] proposed a VAE-based purification method that requires no additional clean data. Qin et al. [34] introduced a benchmark for availability attacks.

# B  Contrastive Learning

Here we introduce Info-NCE-based contrastive learning. As shown in Code 1, strong augmentation is a key component in contrastive learning. Different augmented views of an image focus on various details. If $x, x_{pos}$ are two augmented views of the same image, we say $(x, x_{pos})$ is a positive pair. Conversely, a negative pair $(x, x_{neg})$ contains two augmented views of two different images. Given an encoder $g$, we denote $z = g(x)$ for simplicity and the InfoNCE loss is defined as:

$$\mathcal{L}_{\texttt{InfoNCE}} = \frac{1}{N} \sum_i \frac{s(z_i, z_i')}{\frac{1}{N} \sum_j s(z_i, z_j')},$$

where $\{(z_i, z_i')\}_{i=1}^N$ is a set of features of positive pairs and $\{(z_i, z_j')\}_{j=1}^N$ is a set of features of negative pairs for each $x_i$; the function $s(z, z') = \exp(\frac{z \cdot z'^\top}{T||z||||z'||})$ with a temperature parameter $T$. This object function aims to maximize the cosine similarity of positive pairs while minimizing the cosine similarity of negative pairs.

# C  Experiment Details

## C.1  Datasets and Networks

**CIFAR.** CIFAR-10/CIFAR-100 [26] consists of 50000 training images and 10000 test images in 10/100 classes. All images are $32 \times 32$ colored ones.

**Tiny-ImageNet.** Tiny-ImageNet classification challenge [27] is similar to the classification challenge in the full ImageNet ILSVRC [38]. It contains 200 classes. The training has 500 images for each class and the test set has 100 images for each class. All images are $64 \times 64$ colored ones.

**Mini-ImageNet.** Mini-ImageNet dataset was originally designed for few-shot learning [49]. We modify it for a classification task. The modified dataset contains 100 classes. The training set has 500 images for each class. The test set has 100 images for each class. All images are $84 \times 84$ colored ones.

**ImageNet-100.** ImageNet-100 is a subset of ImageNet-1k Dataset from ImageNet Large Scale Visual Recognition Challenge 2012 [38]. It contains 100 random classes. The training set has 130,000 images. The test set has 5,000 images. Images are processed to 224x224 colored ones as input data to models.

**ResNet.** On CIFAR-10/CIFAR-100, we set the kernel size of the first convolutional layer to 3 and removed the following max-pooling layer. On other datasets, we do not modify the models.

## C.2 Data Augmentation

In Code 1, we show the different implementations of data augmentation between supervised learning and contrastive learning. For supervised learning, we consider the typical augmentations including Crop and HorizontalFlip. For contrastive learning, we consider the typical augmentations including ResizedCrop, HorizontalFlip, ColorJitter, and Grayscale, and its default strength $s = 1$. In the generation process of our AUE and AAP attacks, we replace the supervised augmentations with contrastive-like augmentations of a strength parameter $s$.

Code Listing 1: Different data augmentations used in supervised learning and contrastive learning on CIFAR-10/100 datasets. The intensity of contrastive augmentations can be adjusted via strength $s$.

```
# Supervised augmentations
Compose([RandomCrop(size=32, padding=4), RandomHorizontalFlip(p=0.5),
        ToTensor()])
# Contrastive augmentations
s = 1.0  # Strength is 1.0 by default for contrastive learning.
Compose([RandomResizedCrop(size=32, scale=(1-0.9*s, 1.0)),
        RandomHorizontalFlip(p=0.5),
        RandomApply([ColorJitter(brightness=0.4*s, contrast=0.4*s,
            saturation=0.4*s, hue=0.1*s)], p=0.8*s),
        RandomGrayscale(p=0.2*s), ToTensor()])
```

## C.3 Details of AUE and AAP

We leverage differentiable augmentation modules in Konia[2] [37] which is a differentiable computer vision library for PyTorch. The contrastive augmentations for Tiny/Mini-ImageNet and ImageNet-100 are similar to those for CIFAR-10/100 in Code 1 but only adapt the image size.

**AUE.** We train the reference model for $T = 60$ epochs with SGD optimizer and cosine annealing learning rate scheduler. The batch size of training data is $128$. The initial learning rate $\alpha_\theta$ is $0.1$, weight decay is $10^{-4}$ and momentum is $0.9$. In each epoch, we update the model for $T_\theta = 391$ iterations and update poisons for $T_\delta = 391$ iterations. For ImageNet-100, we set $T_\theta = T_\delta = 1016$. The PGD process for noise generation takes $T_p = 5$ steps with step size $\alpha_\delta = 0.8/255$. The augmentation strength $s = 0.6$ for CIFAR-10 and $s = 1.0$ for CIFAR-100, Tiny-ImageNet, Mini-ImageNet, and ImageNet-100. Additional experiments of the selection of strength parameters are shown in Appendix D.4.

**AAP.** We train the reference model for $T = 40$ epochs, and the initial learning rate $\alpha_\theta$ is $0.5$. The PGD process for noise generation takes $T_p = 250$ steps with step size $\alpha_\delta = 0.08/255$. Other settings are the same as AUE. The label translation is $K = 1$. The augmentation strength $s = 0.4$ for CIFAR-10 and $s = 0.8$ for CIFAR-100, Tiny-ImageNet, Mini-ImageNet, and ImageNet-100.

Besides targeted AP and AAP attacks described in Equation (3) and Algorithm 2, untargeted attacks refer to maximizing the loss between the image and its true label, rather than minimizing the loss between the image and a shifted label. We only report untargeted attack results in Table 3 for CIFAR-10 and discuss them more in Appendix D.5.

**Sample-wise Attack.** When a poisoning map $\delta(\boldsymbol{x}, y)$ only depends on label $y$, the resulting attack is called a class-wise attack; otherwise, it is a sample-wise attack. In this paper, we generate sample-wise attacks.

## C.4 Baseline attacks.

A CP attack implements contrastive error minimization on a specific contrastive learning algorithm. There are three specified attacks including CP-SimCLR, CP-MoCo, and CP-BYOL. Similarly, TUE has three specified attacks including TUE-SimCLR, TUE-MoCo, and TUE-SimSiam. We report the best results of CP attacks and TUE attacks according to the worst-case unlearnability. Besides, we provide detailed results in Appendix D.6 for more discussion.

---

[2]https://github.com/kornia/kornia

### C.5 Evaluation Algorithms

**Contrastive learning.** The setup for SimCLR, MoCo, BYOL, and SimSiam are shown in Table 8. The 100-epoch linear probing stage uses an SGD optimizer and a scheduler that decays 0.2 at 60, 75, and 90 epochs. The probing learning rate is 1.0 for SimCLR, MoCo, BYOL, and 5.0 for SimSiam on CIFAR-10/100, Tiny/Mini-ImageNet. On ImageNet-100, the unsupervised contrastive learning optimizes 200 epochs and the linear probing uses a learning rate of 10.0. Other settings are the same as other datasets. After generating our attacks on CIFAR-10/100, we report average test accuracy after 3 evaluations with random seeds.

**Supervised learning.** We augment the training data by RandomHorizontalFlip and RandomCrop with padding size $l/8$ on CIFAR-10/100 and Tiny/Mini-ImageNet. $l$ is the image size. On ImageNet-100, we augment using RandomResizedCrop and RandomHorizontalFlip.

**SupCL and FixMatch.** We use ResNet-18 for SupCL evaluation on CIFAR-10 and CIFAR-100. For FixMatch evaluation, we use WideResNet-28-2 and 4000 labeled data on CIFAR-10; we use WideResNet-28-8 and 10000 labeled data on CIFAR-100.

Table 8: Details of supervised and contrastive evaluations.

|  | SL | SimCLR | MoCo | BYOL | SimSiam |
|---|---|---|---|---|---|
| Batch size | 512 | 512 | 512 | 512 | 512 |
| Epochs | 200 | 1000 | 1000 | 1000 | 1000 |
| Loss function | CE | InfoNCE | InfoNCE | MSE | Similarity |
| Optimizer | SGD | SGD | SGD | SGD | SGD |
| Learning rate | 0.5 | 0.5 | 0.3 | 1.0 | 0.1 |
| Weight decay | 1e-4 | 1e-4 | 1e-4 | 1e-4 | 1e-4 |
| Momentum | 0.9 | 0.9 | 0.9 | 0.9 | 0.9 |
| Scheduler | Cosine | Cosine | Cosine | Cosine | Cosine |
| Warmup | 10 | 10 | 10 | 10 | 10 |
| Temperature | - | 0.5 | 0.2 | - | - |
| Encoder momentum | - | - | 0.99 | 0.999 | - |

## D  Additional Experiments

### D.1  Computation Consumption

We report the time consumption of generating AUE and AAP attacks. For CIFAR-10/100, Tiny/Mini-ImageNet, experiments are conducted using a single NVIDIA GeForce RTX 3090 GPU. For ImageNet-100, experiments are conducted using a single NVIDIA A800 GPU. On CIFAR-10/100, AUE/AAP costs around 2.7/2.2 hours. On Mini-ImageNet, AUE/AAP costs around 2.5/2 hours. On Tiny-ImageNet, AUE/AAP costs around 2.5/3.8 hours. On ImageNet-100, AUE/AAP costs around 12/10 hours. In comparison, on CIFAR-10/100 and using the same device, CP-SimCLR costs around 48 hours, TUE-MoCo costs around 8.5 hours, and TP costs around 16 hours to generate poisons. Our supervised poisoning attacks are much more efficient than contrastive poisoning attacks.

### D.2  Training Process on Poisoned Data

In Figure 7, we evaluate the training and test accuracy during SL and SimCLR training on poisoned data. In very early epochs where the training underfits the poisoned data, checkpoints from both SL and SimCLR possibly process weak usability. After a few epochs, the test accuracy rapidly goes down to an unusable level. For SimCLR, the accuracy slowly increases in the middle and later stages of training. It aligns with the overall trend of gradually decreasing uniformity gap and relatively stable alignment gap as shown in Figure 5b for AUE.

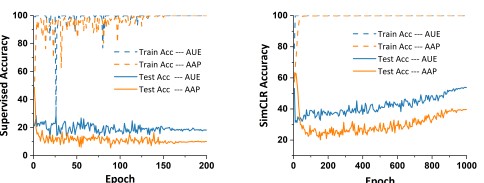

Figure 7: Training process on poisoned CIFAR-10. Left: Supervised learning. Right: SimCLR.

## D.3 Visualization

We scale imperceptible perturbations from [-8/255, 8/255] to [0,1] and show their images in Figure 8. Enhanced data augmentations endow AUE with more complicated patterns than UE. In terms of frequency, they are more high-frequency than UE. Since contrastive augmentations include grayscale that squeezes low-frequency shortcuts [30], attacks against CL first need to come through them and thus prefer high-frequency patterns. Moreover, we check the class-wise separability of perturbations using t-SNE visualization [48] in Figure 8. Perturbations from AUE and T-AAP are less separable than those from UE and T-AP and coincide with the characteristics of perturbations from contrastive error minimization [16].

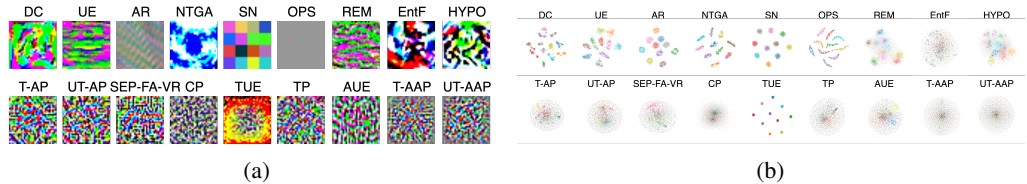

(a)                      (b)

Figure 8: (a) Perturbation images of availability attacks on CIFAR-10. (b) T-SNE visualization of perturbations. In each figure, the top row includes attacks that are not effective against contrastive learning, and the bottom row includes attacks that have contrastive unlearnability.

## D.4 Strength Selection

**AUE.** We gradually increase the data augmentation strength $s$ in the supervised error minimization according to Algorithm 1. In Figure 9a, the SimCLR accuracy prominently decreases as the strength grows, while the supervised learning accuracy slightly increases. Compared to UE, our AUE attacks largely improve contrastive unlearnability while keeping similar supervised unlearnability. On CIFAR-10, too strong strengths might compromise the unlearnability. Thus, we generate our augmented unlearnable example (AUE) attacks taking $s = 0.6$ for CIFAR-10, and $s = 1.0$ for CIFAR-100.

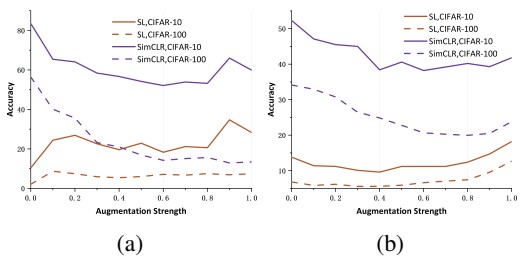

(a)             (b)

Figure 9: (a) Influence of augmentations in AUE. (b) Influence of augmentations in AAP.

**AAP.** We gradually increase the data augmentation strength $s$ in the supervised error maximization according to Algorithm 2. In Figure 9b, the SimCLR accuracy decreases with the strength, while the supervised learning accuracy slightly increases. Proper augmentation strengths improve the contrastive unlearnability but too large $s$ might introduce difficulty in poison generation and harm the supervised unlearnability. We select $s = 0.4$ for CIFAR-10 and $s = 0.8$ for CIFAR-100.

Table 9: Alignment and uniformity gaps of AUE with different strengths.

| Strength | $\mathcal{AG}$ | $\mathcal{UG}$ | Accuracy |
|----------|------|------|----------|
| $s = 0.0$ | 0.14 | 0.07 | 83.5 |
| $s = 0.2$ | 0.21 | 0.24 | 64.1 |
| $s = 0.4$ | 0.25 | 0.28 | 56.7 |
| $s = 0.6$ | 0.27 | 0.34 | 52.4 |

**Strength and Gaps** On CIFAR-10, we gradually increase the augmentation strength from 0 to the default setting, i.e. $s = 0.6$ in the generation of AUE attacks and evaluate the alignment gaps, uniformity gaps, and the SimCLR Accuracy in Table 9. In this case, the larger the gaps, the lower the accuracy of SimCLR.

## D.5 Targeted and Untargeted AAP

Instead of Equation (3), the untargeted attack refers to the following optimization:

$$\max_{\delta} \mathbb{E}_{\mathcal{D}_c} \left[ \mathcal{L}_{\text{SL}}(\boldsymbol{x} + \delta(\boldsymbol{x}, y), y; f^*) \right].$$

In a targeted attack, the perturbations for a class of data are optimized to fit another label, so they finally contain the non-robust feature of the target class. However, there is no consistent target label for a class of data in an untargeted attack. Since availability attacks create shortcuts for the classification task [56], the untargeted attack becomes more difficult than the targeted one when the number of classes increases. [13] also reports that the generation of untargeted attacks is unstable and they focus on targeted attacks on more complex datasets. Thus, we only perform untargeted AAP attacks on the simple dataset CIFAR-10 and report the results in Table 3. As a complement to it, we present the performance of targeted attacks on CIFAR-10 in Table 10, where the untargeted attacks are better than targeted ones in the worst-case unlearnability.

Table 10: Targeted and untargeted AP and AAP attacks on CIFAR-10.

| | Attacks | SL | SimCLR | MoCo | BYOL | SimSiam | Worst |
|---|---|---|---|---|---|---|---|
| AP | Untargeted | 9.6 | 41.5 | 31.5 | 44.0 | 42.8 | 44.0 |
| | Targeted | 9.5 | 48.4 | 53.8 | 53.0 | 51.1 | 53.8 |
| AAP | Untargeted | 29.7 | 32.3 | 23.2 | 35.5 | 34.1 | **35.5** |
| | Targeted | 9.2 | 39.1 | 40.4 | 43.3 | 42.1 | 43.3 |

## D.6 Additional results for CL-based attacks

CP and TUE attacks are based on contrastive error minimization. The poisoning generation depends on a specific contrastive learning algorithm. For example, CP-SimCLR is generated by minimizing the contrastive error of SimCLR. To check the effect of generation algorithm selection on the worst-case unlearnability, we present detailed attack performance of specified CP and TUE attacks on CIFAR-10/100 in Table 11. For CP attacks, only CP-BYOL is effective against supervised learning on CIFAR-10 and no variants work for SL on CIFAR-100. For TUE attacks, the TUE-MoCo is significantly better than other variants on both CIFAR-10 and CIFAR-100.

Table 11: Detailed attack performance of CP and TUE attacks by specifying the underlying algorithm for poisoning generation.

| Attacks | CIFAR-10 | | | | | | CIFAR-100 | | | | | |
|---|---|---|---|---|---|---|---|---|---|---|---|---|
| | SL | SimCLR | MoCo | BYOL | SimSiam | Worst | SL | SimCLR | MoCo | BYOL | SimSiam | Worst |
| CP-SimCLR | 94.5 | 38.7 | 69.3 | 79.5 | 29.2 | 94.5 | 74.7 | 10.5 | 30.7 | 22.6 | 7.7 | 74.7 |
| CP-MoCo | 94.5 | 53.7 | 47.9 | 56.8 | 47.1 | 94.5 | 74.4 | 15.2 | 13.4 | 16.4 | 14.1 | 74.4 |
| CP-BYOL | 11.0 | 39.3 | 32.7 | 41.8 | 37.9 | 41.8 | 74.7 | 29.7 | 35.5 | 35.7 | 29.5 | 74.7 |
| TUE-SimCLR | 10.6 | 48.1 | 71.2 | 79.5 | 39.0 | 79.5 | 1.0 | 16.9 | 36.7 | 40.6 | 7.8 | 40.6 |
| TUE-MoCo | 10.1 | 57.2 | 51.6 | 60.1 | 58.5 | 60.1 | 1.0 | 19.9 | 19.6 | 22.3 | 18.6 | 22.3 |
| TUE-SimSiam | 9.9 | 82.5 | 80.7 | 84.3 | 81.8 | 84.3 | 1.1 | 33.9 | 31.0 | 40.9 | 10.3 | 40.9 |

## D.7 Poisoning Budget

In the main body, we consider the poisoning attacks constrained in a $L_\infty$-norm ball with radius $8/255$. The constraint is to ensure perturbations are imperceptible to human eyes. We investigate the influence of different poisoning budgets. AUE and AAP attacks are generated with poisoning budgets of $2/255, 4/255, 6/255$ and are evaluated by SL and SimCLR. In Table 12, the larger the poisoning budgets, the better the attack performance.

Table 12: Performance(%) of attacks generated with different poisoning budgets on CIFAR-10.

| | Budget | AUE | AAP |
|---|---|---|---|
| SL | 2/255 | 34.5 | 50.7 |
| | 4/255 | 28.5 | 19.7 |
| | 6/255 | 26.8 | 12.3 |
| SimCLR | 2/255 | 84.8 | 87.0 |
| | 4/255 | 70.1 | 66.6 |
| | 6/255 | 59.4 | 51.1 |

Table 13: Performance(%) of attacks with different poisoning ratios on CIFAR-10.

| | Ratio | AUE | AAP |
|---|---|---|---|
| SL | 95% | 75.6 | 82.1 |
| | 90% | 82.2 | 86.6 |
| | 80% | 87.6 | 89.8 |
| SimCLR | 95% | 69.7 | 76.8 |
| | 90% | 74.5 | 82.1 |
| | 80% | 79.7 | 85.5 |

**D.8  Poisoning Ratio**

Availability attacks are sensitive to the proportion of poisoned data in the dataset and usually need to poison the whole dataset [22, 13]. In the main body, we report results when the poisoning ratio is $100\%$. Here, we investigate the influence of the poisoning ratio on the attack performance of AUE and AAP. Table 13 illustrates that our augmented methods inherit the vulnerability to poisoning ratio from basic approaches, i.e. UE and AP, though AUE is more robust than AAP. This characteristic also necessitates the prompt processing of newly acquired clean data, imposing higher efficiency demands on the generation of attacks.

**D.9  Defense**

On the defense side against availability attacks, AT [31] and AdvCL [25]) applied adversarial training in supervised learning and contrastive learning respectively; ISS [30] and UEraser [35] leveraged designed data augmentations to eliminate supervised unlearnability; AVATAR [9] employed a diffusion model to purify poisoned data. In Table 14, we evaluate our attacks through these defense methods as well as SimCLR with Cutout [8], Random noise, and Gaussian Blur. The defensive budget for AT and AdvCL is $8/255$; the length parameter for Couout is 8; the kernel size for Gaussian Blur is 3; the variance for Random noise is 8/255.

The defense performance of a method differs when facing different attacks. For example, UEraser can recover the accuracy of TUE-SimCLR from $10.6\%$ to above $92.7\%$, while its effect on our AUE attack is much weaker. At the cost of a significant amount of extra training time, adversarial training, i.e. AT and AdvCL, can increase accuracy to around $80\%$. ISS mitigates the supervised unlearnability of evaluated attacks back to levels close to $85\%$, but its Grayscale component may even have negative effects. Gaussian Blur is more effective than Cutout and Random noise for contrastive learning.

Recently, diffusion models have provided a powerful tool to purify image perturbations [55, 32, 43]. Here we evaluate AVATAR which employs a diffusion model trained on the CIFAR-10 training dataset. From the table, AVATAR generally achieves the best defense against our proposed attacks, but the final accuracy still exhibits a gap compared to training with clean data. We believe it's an interesting and worthy future direction to improve attacks' resilience to potential defenses while maintaining algorithm transferability

Table 14: Performance(%) under defenses on CIFAR-10. Here TUE is based on SimCLR.

|  | Defense | AUE | AAP | AP | TUE |
|---|---|---|---|---|---|
|  | No Defense | 18.9 | 9.2 | 9.5 | 10.6 |
|  | UEraser | 63.2 | 64.7 | 68.0 | 92.7 |
|  | -Lite | 60.6 | 66.8 | 70.7 | 92.2 |
|  | -Max | 72.8 | 79.5 | 80.2 | 93.2 |
| SL | ISS | 82.6 | 82.3 | 81.7 | 82.7 |
|  | -Grayscale | 18.2 | 9.1 | 11.4 | 28.0 |
|  | -JPEG | 84.9 | 84.3 | 84.6 | 82.1 |
|  | AVATAR | 85.0 | 88.0 | 87.7 | 83.2 |
|  | AT | 83.8 | 81.6 | 81.0 | 81.7 |
|  | No Defense | 52.4 | 39.1 | 48.4 | 48.1 |
|  | Cutout | 51.8 | 37.9 | 49.2 | 49.6 |
| SimCLR | Random Noise | 60.5 | 62.4 | 66.4 | 70.0 |
|  | Gaussian Blur | 69.1 | 76.7 | 75.5 | 79.3 |
|  | AVATAR | 83.1 | 80.8 | 81.1 | 83.0 |
|  | AdvCL | 80.9 | 78.4 | 77.5 | 80.1 |

Figure 10: Clean and poisoned linear probing on CIFAR-10.

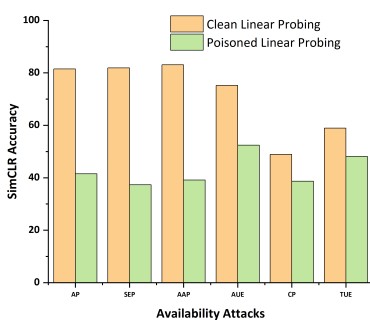

**D.10  Discussion of Clean Linear Probing**

While our threat model linear probes on poisoned data, He et al. [16] use clean data for linear probing instead. In Figure 10, we compare the final classification performance of SimCLR models in these two settings. Feature extractors are trained on poisoned data and are fixed. We focus on the classification performance after linear probing on clean or poisoned data. While CP and TUE obtain similar attack performance in both cases, clean linear probing can mitigate SL-based attacks including AP, SEP-FA-VR, AAP, and AUE. On one hand, for SL-based poisoning, the dissimilarities between clean features and poisoned features hinder a classifier head obtained by poisoned linear probing in generalizing to clean features, as discussed in Section 3.2. However, clean features still

contain useful information and can derive another classifier head to perform classification. On the other hand, contrastive error-minimizing noises confuse the feature extractor directly such that even clean data fail to activate useful features for classification. But in general, given a responsible data publisher who protects data using availability attacks before release, an unauthorized data collector has no access to unprocessed data for clean linear probing. Thus, it is sufficient to achieve contrastive unlearnability with poisoned linear probing in real scenarios.

# E   Toy Example

We study a model $f = h \circ g : \mathbb{R}^d \to \mathbb{R}^n$ with a normalized feature extractor $g : \mathbb{R}^d \to \mathbb{R}^n$ such that $||g(\boldsymbol{x})|| \equiv 1$ and a full rank linear classifier $h : \mathbb{R}^n \to \mathbb{R}^n$ in the sense that $h(\boldsymbol{z}) = W\boldsymbol{z} + \boldsymbol{b}$ with a full rank square matrix $W \in \mathbb{R}^{n \times n}$. By singular values decomposition (SVD), $W = U\Sigma V$ with orthogonal matrices $U, V \in \mathbb{R}^{n \times n}$ and $\Sigma = \mathrm{diag}(\sigma_1, \cdots, \sigma_n), \sigma_1 \geq \cdots \geq \sigma_n > 0$. Let $\mathcal{D}$ be a balanced data distribution, i.e. each class would be sampled with the same probability, $\mathcal{D}_x$ be the margin distribution, and $\mu$ be an augmentation distribution. Assume the supervised loss $\mathcal{L}_{\mathrm{SL}}$ is the mean squared error, and the contrastive loss $\mathcal{L}_{\mathrm{CL}}$ contains only one negative example:

$$\mathcal{L}_{\mathrm{SL}}(\boldsymbol{x}, y, \pi) = \frac{1}{n}||h \circ g(\pi(\boldsymbol{x})) - \boldsymbol{e}_y||^2$$

$$\mathcal{L}_{\mathrm{CL}}(\boldsymbol{x}, \boldsymbol{x}^-, \pi, \tau, \rho) = \log(1 + \frac{e^{g(\pi(\boldsymbol{x}))^\top g(\rho(\boldsymbol{x}))}}{e^{g(\pi(\boldsymbol{x}))^\top g(\tau(\boldsymbol{x}^-))}}).$$

**Proposition E.1.** *Let $\mathcal{E}_{\mathrm{SL}} = \mathbb{E}_{\mathcal{D},\mu}\left[\mathcal{L}_{\mathrm{SL}}(\boldsymbol{x}, y, \pi)\right]$. With probability at least $1 - 4\sqrt{\mathcal{E}_{\mathrm{SL}}}$, it holds*

$$\mathcal{L}_{\mathrm{CL}}(\boldsymbol{x}, \boldsymbol{x}^-, \pi, \tau, \rho) < \frac{1}{n}\log(1 + \frac{\sigma_n}{\sigma_n - 2n\sqrt{\mathcal{E}_{\mathrm{SL}}}})$$

$$+ \frac{n-1}{n}\log(1 + \frac{\sigma_1^2\sigma_n - \sigma_n(1 - \sqrt{2n\sqrt{\mathcal{E}_{\mathrm{SL}}}})^2}{\sigma_1^2\sigma_n - 2n\sigma_1^2\sqrt{\mathcal{E}_{\mathrm{SL}}}}).$$

**Remark E.2.** *1) Assumptions of a square matrix and positive singular values are necessary. Otherwise, the dimensional reduction of feature space impairs the relation between supervised and contrastive losses. 2) Since supervised losses contain limited information about negative pairs, this inequality is naturally loose. However, in the case that supervised learning fits very well, it at least implies that positive features $g(\tau(\boldsymbol{x}))$ are closer to $g(\pi(\boldsymbol{x}))$ than negative features $g(\rho(\boldsymbol{x}^-))$.*

## E.1   Lemmas

**Lemma E.3.** *For any $\boldsymbol{z} \in \mathbb{R}^n$,*

$$\sigma_n||\boldsymbol{z}|| \leq ||W\boldsymbol{z}|| \leq \sigma_1||\boldsymbol{z}||.$$

*Proof.* Denote $\tilde{\boldsymbol{z}} = (\tilde{z}_1, \cdots, \tilde{z}_n)^\top = V\boldsymbol{z}$. Since orthogonal matrices preserve the norm,

$$||W\boldsymbol{z}|| = ||U\Sigma V\boldsymbol{z}|| = ||\Sigma\tilde{\boldsymbol{z}}|| = \sqrt{\sum_{i=1}^n \sigma_i^2 \tilde{z}_i^2},$$

$$\sigma_n||\boldsymbol{z}|| = \sigma_n||\tilde{\boldsymbol{z}}|| \leq \sqrt{\sum_{i=1}^n \sigma_i^2 \tilde{z}_i^2} \leq \sigma_1||\tilde{\boldsymbol{z}}|| = \sigma_1||\boldsymbol{z}||.$$

$\square$

**Lemma E.4.** *If $\mathcal{E}_{\mathrm{SL}} \leq \epsilon$, then with probability at least $1 - \sqrt{\epsilon}$*

$$||h \circ g(\pi(\boldsymbol{x})) - \boldsymbol{e}_y|| < \sqrt{n\sqrt{\epsilon}},$$

*where $(\boldsymbol{x}, y) \sim \mathcal{D}, \pi \sim \mu$.*

*Proof.* As

$$\mathcal{E}_{\text{SL}} = \mathbb{E}_{\substack{(\boldsymbol{x},y)\sim\mathcal{D} \\ \pi\sim\mu}} \left[\frac{1}{n}||h \circ g(\pi(\boldsymbol{x})) - \boldsymbol{e}_y||^2\right],$$

by Markov's inequality, it has

$$\Pr(\frac{1}{n}||h \circ g(\pi(\boldsymbol{x})) - \boldsymbol{e}_y||^2 \geq \sqrt{\epsilon}) \leq \sqrt{\epsilon}.$$

$\square$

**Lemma E.5.** *If $\mathcal{E}_{\text{SL}} \leq \epsilon$, then with probability at least $1 - 2\sqrt{\epsilon}$*

$$g(\pi(\boldsymbol{x}))^\top g(\tau(\boldsymbol{x})) > 1 - \frac{2n\sqrt{\epsilon}}{\sigma_n},$$

*where $\boldsymbol{x} \sim \mathcal{D}_{\boldsymbol{x}}, \pi, \tau \sim \mu$.*

*Proof.* By Lemma E.4, with probability at least $1 - 2\sqrt{\epsilon}$,

$$||h \circ g(\pi(\boldsymbol{x})) - \boldsymbol{e}_y|| < \sqrt{n\sqrt{\epsilon}} \quad \text{and} \quad ||h \circ g(\tau(\boldsymbol{x})) - \boldsymbol{e}_y|| < \sqrt{n\sqrt{\epsilon}}.$$

By the triangle inequality,

$$||h \circ g(\pi(\boldsymbol{x})) - h \circ g(\tau(\boldsymbol{x}))|| < 2\sqrt{n\sqrt{\epsilon}}$$

Since $g$ is normalized, by Lemma E.3 we have

$$\begin{aligned}
g(\pi(\boldsymbol{x}))^\top g(\tau(\boldsymbol{x})) &= 1 - \frac{1}{2}||g(\pi(\boldsymbol{x})) - g(\tau(\boldsymbol{x}))||^2 \\
&\geq 1 - \frac{1}{2\sigma_n^2}||h \circ g(\pi(\boldsymbol{x})) - h \circ g(\tau(\boldsymbol{x}))||^2 \\
&> 1 - \frac{2n\sqrt{\epsilon}}{\sigma_n}.
\end{aligned}$$

$\square$

**Lemma E.6.** *Assume $\mathcal{D}$ is a balanced dataset. If $\mathcal{E}_{\text{SL}} \leq \epsilon$, then with probability at least $1 - 2\sqrt{\epsilon}$, one of the following two conditions holds*

1. *with probability $\frac{n-1}{n}$,*

$$g(\pi(\boldsymbol{x}))^\top g(\tau(\boldsymbol{x}^-)) < 1 - \frac{(1 - \sqrt{2n\sqrt{\epsilon}})^2}{\sigma_1^2};$$

2. *with probability $\frac{1}{n}$,*

$$g(\pi(\boldsymbol{x}))^\top g(\tau(\boldsymbol{x}^-)) \leq 1.$$

*Proof.* 1. With probability $\frac{n-1}{n}$, for $(\boldsymbol{x}, y), (\boldsymbol{x}^-, y^-) \sim \mathcal{D}, y \neq y^-$. By Lemma E.4, with probability at least $1 - 2\sqrt{\epsilon}$,

$$||h \circ g(\pi(\boldsymbol{x})) - \boldsymbol{e}_y|| < \sqrt{n\sqrt{\epsilon}} \quad \text{and} \quad ||h \circ g(\tau(\boldsymbol{x}^-)) - \boldsymbol{e}_{y^-}|| < \sqrt{n\sqrt{\epsilon}}.$$

By the triangle inequality,

$$\begin{aligned}
||g(\pi(\boldsymbol{x})) - g(\tau(\boldsymbol{x}^-))|| &\geq \frac{1}{\sigma_1}||h \circ g(\pi(\boldsymbol{x})) - h \circ g(\tau(\boldsymbol{x}^-))|| \\
&\geq \frac{1}{\sigma_1}(||\boldsymbol{e}_y - \boldsymbol{e}_{y^-}|| - ||h \circ g(\pi(\boldsymbol{x})) - \boldsymbol{e}_y|| - ||h \circ g(\tau(\boldsymbol{x}^-)) - \boldsymbol{e}_{y^-}||) \\
&> \frac{\sqrt{2} - 2\sqrt{n\sqrt{\epsilon}}}{\sigma_1}.
\end{aligned}$$

Since $g$ is normalized,

$$g(\pi(\boldsymbol{x}))^\top g(\tau(\boldsymbol{x}^-)) = 1 - \frac{1}{2}||g(\pi(\boldsymbol{x})) - g(\tau(\boldsymbol{x}^-))||^2$$

$$< 1 - \frac{(1 - \sqrt{2n\sqrt{\epsilon}})^2}{\sigma_1^2}.$$

2. As we assume $\mathcal{D}$ is a balanced dataset, with probability $\frac{1}{n}$, for $(\boldsymbol{x}, y), (\boldsymbol{x}^-, y^-) \sim \mathcal{D}$, $y = y^-$. Since $g$ is normalized,

$$g(\pi(\boldsymbol{x}))^\top g(\tau(\boldsymbol{x}^-)) = 1 - \frac{1}{2}||g(\pi(\boldsymbol{x})) - g(\tau(\boldsymbol{x}^-))||^2$$

$$\leq 1 - \frac{1}{2\sigma_1^2}||h \circ g(\pi(\boldsymbol{x})) - h \circ g(\tau(\boldsymbol{x}^-))||^2$$

$$\leq 1.$$

$\square$

## E.2 Proof of Proposition E.1

*Proof.* Let $\mathcal{E}_{\mathrm{SL}} = \epsilon$. Combining Lemma E.5 and Lemma E.6, for a sample $\boldsymbol{x}$ and its negative sample $\boldsymbol{x}^-$ i.i.d from $\mathcal{D}_{\boldsymbol{x}}$, and data augmentation method $\pi, \tau, \rho$ i.i.d from $\mu$, with probability at least $1 - 4\sqrt{\mathcal{E}_{\mathrm{SL}}}$, it holds that

$$\mathcal{L}_{\mathrm{CL}}(x, x^-, \pi, \tau, \rho) = -\log \frac{e^{g(\pi(\boldsymbol{x}))^\top g(\tau(\boldsymbol{x}))}}{e^{g(\pi(\boldsymbol{x}))^\top g(\tau(\boldsymbol{x}))} + e^{g(\pi(\boldsymbol{x}))^\top g(\rho(\boldsymbol{x}^-))}}$$

$$= \log(1 + \frac{e^{g(\pi(\boldsymbol{x}))^\top g(\rho(\boldsymbol{x}^-))}}{e^{g(\pi(\boldsymbol{x}))^\top g(\tau(\boldsymbol{x}))}})$$

$$< \frac{n-1}{n}\log(1 + \frac{1 - \frac{(1 - \sqrt{2n\sqrt{\mathcal{E}_{\mathrm{SL}}}})^2}{\sigma_1^2}}{1 - \frac{2n\sqrt{\mathcal{E}_{\mathrm{SL}}}}{\sigma_n}}) + \frac{1}{n}\log(1 + \frac{1}{1 - \frac{2n\sqrt{\mathcal{E}_{\mathrm{SL}}}}{\sigma_n}})$$

$$= \frac{1}{n}\log(1 + \frac{\sigma_n}{\sigma_n - 2n\sqrt{\mathcal{E}_{\mathrm{SL}}}}) + \frac{n-1}{n}\log(1 + \frac{\sigma_1^2\sigma_n - \sigma_n(1 - \sqrt{2n\sqrt{\mathcal{E}_{\mathrm{SL}}}})^2}{\sigma_1^2\sigma_n - 2n\sigma_1^2\sqrt{\mathcal{E}_{\mathrm{SL}}}}).$$

$\square$

