# OpenReview forum: "Efficient Availability Attacks against Supervised and Contrastive Learning Simultaneously"
_NeurIPS.cc/2024/Conference — NeurIPS 2024 poster_

### Official Review · Reviewer_vjzY · 2024-07-08

**Soundness:** 2
**Presentation:** 3
**Contribution:** 3
**Rating:** 6
**Confidence:** 4

**Summary:**

This paper studies the transferability of unlearnable examples (UEs) across different learning paradigms, i.e., supervised learning and self-supervised contrastive learning. Different from existing works including both supervised UE generation methods and hybrid methods, this paper argues that strong data augmentations with supervised learning are sufficient for unsupervised unlearnability. It then proposes two strong data augmentations equipped UE generation methods AUE (with standard training) and AAP (with adversarial training). The two proposed methods demonstrate certain superiority over the existing methods on low- and high-resolution datasets.

**Strengths:**

1. The proposed methods are extremely efficient than contrastive learning involving UE generation methods. It leverages supervised learning with strong data augmentation to achieve unsupervised unlearnability against contrastive learning.

2. The proposed method is simple but effective. The revealed finding that strong data augmentation is important for contrastive learning is interesting.

3. Extensive experiments confirmed the capability and efficiency of the proposed attacks.

**Weaknesses:**

1. The alignment and uniformity metrics are all from existing works, which lowers the novelty of the finding in Table 1: "contrastive unlearnability is highly related to huge alignment and uniformity gaps". And the two proposed methods have little to do with the two gaps?

2. While the finding that "supervised learning with strong data augmentation can achieve unsupervised unlearnability" is interesting, the two proposed methods have limited technical novelty. In other words, they do not explore the power of data augmentation more systematically. The current mechanism is very ad-hoc.

3. The performance difference (AUE vs AAP) between low- vs high-resolution datasets is intriguing. AAP targets adversarial training exploitation, so why it should perform better or worse than AUE against contrastive learning? Does contrastive learning have anything to do with adversarial training?

**Questions:**

1. Should the worst-case unlearnability contain other learning paradigms rather than just supervised learning and contrastive learning? For example, MAE or autoregressive pretraining. We cannot enumerate all training paradigms.

2. Why data augmentation is so important for obtaining unsupervised unlearnability, what if there are non-data-augmention-based unsupervised learning methods?

--------
Post rebuttal: I have increased my rating according to the authors' responses.

**Limitations:**

The technical novelty of this work is somewhat limited.

---

> ### Author Rebuttal · Authors · 2024-08-06
>
> We are grateful to the reviewers for their detailed and thoughtful reviews!
> We aim to resolve the issues highlighted and believe our answers will do so.
>
>
> **[Weakness1] Alignment and uniformity gaps**
> - Alignment and uniformity are widely adopted illustrations for the mechanism of contrastive learning. In our problem settings, although the final effectiveness of availability attacks is evaluated through classification accuracy, we still need to carefully examine how the attacks affect contrastive learning at the encoder level. Thus, we borrow the concepts of alignment and uniformity and introduce the gaps between them on clean and poisoned data respectively.
> By studying the gaps, we explain why some attacks are effective against CL algorithms while others are not.
>
> - Effective attacks cause significant differences between the features of poisoned samples and those of clean samples, ultimately rendering the learned classifier ineffective on the ground truth distribution.
>
> - Regarding our proposed methods, Figure 5(b) demonstrates that AUE exhibits significant alignment and uniformity gaps, whereas unenhanced UE does not, as shown in Table 1.
>
>
>
> **[W2] Systematical study on data augmentation** The review said our proposed method is ad-hoc. We believe some misunderstandings need to be clarified.
>
> - *[Technique novelty]* To obtain unlearnability for CL, we are the first to adopt contrastive augmentation in the SL-based training process. On this topic, we have moved away from the path dependence on contrastive error minimization.
>
> - *[Theoretical results]* We provide a toy model in the appendix to theoretically verify the intuition of our method (Proposition E.1): if the supervised training process uses the same data augmentation as CL, then this process also optimizes the contrastive loss.
>
> - *[Augmentation strategies]* In the discussion with Reviewer 5WZD, we explore more data augmentation strategies in our proposed attacks, including augmentation decoupling, augmentation differentiability, and dynamic strategies.
>
> **[W3] AAP better or worse than AUE**
> - On the one hand, when the dataset becomes higher-resolution and more category, the optimization difficulty in generating adversarial poisoning from scratch increases since AP finds non-robust features that rely on a good classifier.
>
> - On the other hand, since supervised error minimization is easy to optimize, the high resolution provides more room for AUE to create effective poisoning patterns to fool the subsequent algorithms.
>
> **[W4] Contrastive learning and adversarial training**
> If we understand correctly, the reviewer's question is why adversarial poisoning (AP) is effective for contrastive learning. We have discussed it in Appendix D.3 of our paper. On the one hand,  [1] shows that some contrastive augmentation can squeeze low-frequency "shortcuts" in unlearnable examples.
> On the other hand, contrastive error minimization-based attacks such as CP and TP generate high-frequency perturbations (see Figure 8 in our paper).
> From the visualization, perturbations from AP contain more high-frequency patterns, which may deceive contrastive learning more easily.
>
> [1] Liu Z, et al. Image shortcut squeezing: Countering perturbative availability poisons with compression. ICML 2023.
>
>
> **[Q1] Learning paradigms other than contrastive learning**
>
> - In Table 5 of our paper, we have checked the supervised contrastive learning (SupCL) and a semi-supervised method FixMatch.
>
> - We evaluate our attacks against MAE by end-to-end fine-tuning on CIFAR-10. In the presence of AUE/AAP, the test accuracy of fine-tuning decreases by 57%/37%. In Figure 2 of the additional document, the training accuracy in the presence of attacks increases more quickly than that in the clean case, while the test accuracy is the opposite, meaning that the attacks create shortcuts for MAE.
> Although our methods were not specifically designed for MAE, they can transfer to MAE.
>
>     |Clean | AUE | AAP |
>     | --- | --- | --- |
>     | 89.63 | 32.36 | 51.93 |
>
>
>
>
> **[Q2] Importance of data augmentation.**
>
> - Data augmentation is an essential component for CL since it creates different views of the same data point, which is fundamental for learning useful representations. Our work leverages stronger data augmentation to improve the effectiveness of SL-based attacks against CL.
>
> - As other reviewers pointed out, there is a special case in CL, i.e., Text-Image contrastive learning algorithm CLIP. The image encoder of CLIP does not involve contrastive augmentation. We conduct experiments on linear probing upon the CLIP encoder and show that our attacks take effect on this algorithm. See our discussion with Reviewer 61jU for detailed results.

---

> > ### Comment · Reviewer_vjzY · 2024-08-08
> > **Thanks for the clarification**
> >
> > Thanks. Most of my concerns have been addressed, thus I increase my rating to 6.

---

### Official Review · Reviewer_J4Hn · 2024-07-12

**Soundness:** 3
**Presentation:** 3
**Contribution:** 2
**Rating:** 6
**Confidence:** 3

**Summary:**

This paper explores efficient availability attacks that target both supervised and contrastive learning algorithms simultaneously. The authors propose two attack methods, named AUE (Augmented Unlearnable Examples) and AAP (Augmented Adversarial Poisoning), which utilize enhanced data augmentations to craft effective poisoning attacks. These methods achieve state-of-the-art unlearnability with reduced computational demands, showing promise for practical applications. The paper evaluates the effectiveness of these attacks across multiple datasets and demonstrates their superiority over existing methods in achieving worst-case unlearnability while handling high-resolution datasets efficiently.

**Strengths:**

- The motivation behind the study is sound, as it addresses an important gap in existing methods, which are unable to achieve unlearnability for both supervised and contrastive learning. This realization underscores the need to explore techniques that can simultaneously achieve supervised and contrastive unlearnability.
- The two proposed attack methods, AAP (Augmented Adversarial Poisoning) and AUE (Augmented Unlearnable Examples), are presented as simple yet effective.
- The experiments conducted are comprehensive and robust.

**Weaknesses:**

- While the motivation is justified, the technical contributions could be considered direct. Integrating data augmentation into the generation of poisons is a straightforward approach, which may not reflect a non-trivial technical contribution.
- The methods lack theoretical analysis, which raises questions about the guaranteed effectiveness of data augmentation in all scenarios. Additionally, there is no explanation of why data augmentation works effectively or if it completely addresses the issue at hand through augmentation alone.

**Questions:**

What is the biggest challenge in this work?

---

> ### Author Rebuttal · Authors · 2024-08-06
>
> We would like to thank the reviewers for their valuable comments and suggestions! We hope our responses sufficiently address the concerns raised.
>
> **[Weakness1] Theoretical guarantees and why the enhancing data augmentation works.**
>
> - In fact, we conduct theoretical analysis for a toy model in the appendix to verify the intuition of our proposed method (**Proposition E.1**): if the supervised training process uses the same data augmentation as CL, then this process also optimizes an upper bound of the contrastive loss. As a result, supervised error minimization mimics contrastive error minimization. Such mimicry makes the modified SL-based generation process enhance the contrastive unlearnability of final perturbations.
>
> - By the way, we agree with the reviewer that availability attacks currently lack theoretical guarantees similar to those provided by certified adversarial robustness or differential privacy. Although some works have clarified the optimization objectives of such attacks, it is still insufficient to guarantee the final achievement of unlearnability.
>
> **[W2] Technical contribution** To obtain unlearnability for CL, we are the first to adopt contrastive augmentation in the SL-based training process. The proposed method is simple, effective, and efficient. Compared to other more complex algorithms, it is easier to scale to large datasets and has greater potential for real-world applications.
>
> **[Question] Biggest challenge**
> In our opinion, the biggest challenge in this work is breaking free from the reliance on existing paths. Current methods on this topic, including CP, TUE, and TP, are all based on the contrastive training process which is time-consuming and meets challenges in scaling up to real-world datasets. We believe it’s possible to obtain effective availability attacks through the supervised training process which is more efficient and easier to optimize. The remaining task is to consider how to connect the characteristics of contrastive learning and supervised learning and propose an effective algorithm.

---

> > ### Author Response · Authors · 2024-08-13
> >
> > Dear Reviewer J4Hn,
> >
> > We have submitted a rebuttal addressing your comments. Could you please review it and confirm if your concerns have been resolved?
> >
> > Best regards,\
> > Authors

---

> ### Comment · Reviewer_J4Hn · 2024-08-14
>
> I appreciate the authors' response. My concerns regarding the theoretical analysis and technical contributions have been mostly addressed. I would like to raise my score accordingly.

---

### Official Review · Reviewer_5WZD · 2024-07-12

**Soundness:** 3
**Presentation:** 3
**Contribution:** 2
**Rating:** 6
**Confidence:** 4

**Summary:**

This paper claims to introduce a threat scenario where the adversary may use contrastive learning to bypass the unlearnable examples that are crafted only for supervised learning. In this threat model, the authors showcase that previous works on unlearnable examples crafted only for supervised learning may fall short of contrastive learning. Based on this observation, the authors propose a new approach that considers both supervised and contrastive learning when generating unlearnable examples. The proposed approach is also efficient due to the augmentation adopted augmentation strategies. Extensive experiments show that the proposed method is effective in this new threat scenario.

**Strengths:**

The paper is clearly presented. The proposed approach is efficient thanks to the augmentation strategies that imitate the CL. As a consequence, AUE and AAP are also easily scalable to practical datasets, like ImageNet subsets.

The authors provided a very complete analysis to evaluate the proposed methods, including discussions on limitations and performance under different defenses. It provides a relatively complete understanding of the proposed approaches in different threat scenarios.

**Weaknesses:**

The introduced topic is interesting but also limited. The authors point out that there is a lack of unlearnable approaches that are effective against both supervised learning (SL) and contrastive learning (CL). This observation is not fully novel, since CP, TP, and TUE have similar observations and are effective in most cases against both types of learning. For example, CP is an approach that is designed for contrastive learning but is also effective against supervised learning. I would suggest that the author could tone down the claims that hint at this observation as a novel finding.

The main contribution of this work is to employ both SL and CL losses in the generation of unlearnable examples, including augmentations that imitate the CL in the SL scenario. The proposed approaches show better average and worst-case performance, but the improvement is incremental, according to Table 3 and Table 4. I would suggest that the authors could provide a deeper and more detailed analysis of the difference between SL and CL unlearnable examples by conducting more ablation studies on the augmentation strategies.  The working mechanism behind the augmentation strategies is of interest to the research community.

**Questions:**

Please motivate the threat scenario where the proposed unlearnable examples are necessary. Further analysis of the augmentation strategies could be clarified.

**Limitations:**

The authors have discussed most limitations adequately.

---

> ### Author Rebuttal · Authors · 2024-08-06
>
> We thank the reviewers for their constructive feedback and valuable suggestions! We hope that our clarifications address your queries and concerns effectively.
>
>
> **[Weakness1] Topic novelty and previous works**
>
> - As the reviewer mentioned, the CP and TUE papers were the first to point out the vulnerability of unlearnable examples to contrastive learning.
>
> - However, we notice that CL-based attacks are time-consuming and suffer the vulnerability of algorithm transferability, for example, CP is not consistently effective for SL and TP's performance decreases against different evaluation CL algorithms (see Section 5.2 in our paper).
>
> - Regarding the challenges in efficiency and effectiveness, our work provides two SL-based attacks that achieve both SL and CL unlearnability.
>
> - In summary, we sincerely accept your suggestions and will adopt a more modest tone in the abstract and introduction versions and will clarify our contributions in the subsequent.
>
>
> **[W2] Augmentation strategies** We indeed conducted more ablation studies on augmentation strategies. Here are our results.
>
>
>
>
> - **(1)Dynamic augmentation scheme.** In the paper, we implement the proposed attacks with a constant augmentation strength. We also attempt dynamic augmentation schemes including annealing and tempering which linearly change augmentation strength during the perturbation generation process. In Figure 1 of the additional document, the annealing scheme achieves better supervised unlearnability than the tempering scheme; regarding contrastive unlearnability, annealing outperforms tempering for small final strengths and just the opposite for large final strengths. Compared with the constant scheme in the paper, the tempering scheme with a particular final strength is marginally better, while other options are worse. In summary, these two dynamic schemes do not outperform the default constant scheme used in the paper.
>
> - **(2) Applying perturbation before/after augmentation**
> Another key factor in the effectiveness of our method is that perturbation should be added **before** augmentation. In the perturbation generation process, if the perturbation is added before augmentation $\mu$, then its gradient propagates through $\mu$, i.e., $\nabla_\delta f(\mu(x+\delta))$; otherwise, its gradient does not go through $\mu$, i.e., $\nabla_\delta f(\mu(x)+\delta)$.
> We conduct an ablation study of applying perturbation **after** augmentation for AUE in Figure 3 of the additional document. In that case, enhancing augmentation strength does not improve the effectiveness against contrastive learning.
>
> **[Question1] Scenario**
> Let’s continue the scenario described in [1]: Given that facial recognition tools can be trained by scraping people’s photos from the public web, can people still feel secure uploading their selfies to social media?
> Availability attacks (unlearnable examples) provide a tool that allows AI systems to ignore a person’s selfies; however, this is limited to AI systems based on supervised learning algorithms.
> Considering contrastive learning algorithms can currently achieve performance comparable to that of supervised learning, once AI systems begin to use contrastive learning, the protection of selfies collapses.
> This is why the effectiveness of attacks against contrastive learning in real-world scenarios is important.
> Our work provides an efficient method that is effective for both supervised learning and contrastive learning.
>
> [1] Will Douglas. How to stop AI from recognizing your face in selfies. MIT Technology Review. https://www.technologyreview.com/2021/05/05/1024613/stop-ai-recognizing-your-face-selfies-machine-learning-facial-recognition-clearview/
>
> **[Q2] Augmentation strategies** See our reply to [W2]

---

> > ### Comment · Reviewer_5WZD · 2024-08-09
> >
> > Thank the authors for your detailed clarifications and additional analysis. I appreciate all of your efforts. I still have one concern regarding the "enhanced augmentations". In Figure 1 (b) of the rebuttal, does it suggest that both annealing and tempering are not critical for the unlearnable CL? Could you please further motivate your choices? To make it more specific, in line209 of the draft: "In other words, supervised error-minimizing noises with enhanced data augmentations can partially replace the functionality of contrastive error-minimizing noises to deceive contrastive learning." Could you further clarify the enhanced augmentation and motivate it?

---

> > > ### Author Response · Authors · 2024-08-10
> > > **Further clarification**
> > >
> > > Thanks for your response to our rebuttal! We will further clarify our method and the motivation of it in detail.
> > >
> > > **[1] Motivation of augmentation enhancement**
> > > To make clear our motivation, we first revisit the mechanism of error-minimization and then clarify the motivation and the statement in Line 209.
> > >
> > > - **Mechanism of error-minimization** In Equations (2,4) of our paper, error-minimization refers to a min-min optimization of an objective (SL or CL) loss with respect to weights and perturbations. It aims to generate perturbed data that can be extremely easily optimized by the cross-entropy loss for SL or by InfoNCE loss for CL.
> > >
> > >     As a result, this characteristic of poisoned data, which makes the loss function converge easily, leads to a model that only learns the fraudulent synthetic patterns from perturbations and fails to learn the true ground-truth data distribution during training. For example, in Figure 7 of our paper, we closely examine the training process on poisoned data and find it converging rapidly.
> > >
> > > - **Motivation of our method and the statement in Line 209**
> > > To obtain unlearnability for CL, the existing path is contrastive error-minimization (CP, TUE, TP) based on the CL training process.
> > > In this paper, we aim to provide a solution based on the SL training process which is more efficient.
> > >
> > >     Recall that Equation (2) differs from Equation (4) with the loss function choice, say, SL loss vs. CL loss. Our **key insight** is, that using contrastive augmentation in SL loss and optimizing it can implicitly reduce the CL loss to some extent, as shown in Figure 4. (More empirical observations and theoretical analysis about this have been discussed in Section 4.1 and Appendix E.)
> > >
> > >     Thus, for our proposed AUE attack, while it appears to be performing supervised error minimization, it is also carrying out contrastive error minimization.
> > >     That is, as stated in **Line 209**, the enhanced data augmentation allows supervised error-minimization to partially serve the role of contrastive error-minimization. We verify this claim in Figure 5(a) of our paper, in which the InfoNCE loss during training on AUE poisoned data is largely reduced, achieving a similar effect of contrastive error-minimization.
> > >
> > > **[2] Implementation schemes of augmentation in our attacks**
> > >
> > > - **Augmentations in SL and CL** CL typically uses more and stronger data augmentations compared to SL. Specifically, SL generally relies on simpler augmentations like cropping and flipping, whereas contrastive learning incorporates more advanced techniques such as color jittering, grayscale, and others.
> > > SL-based attacks typically leverage the mild data augmentation used in supervised learning.
> > >
> > >
> > > - **Default constant strength scheme**
> > > To enhance the contrastive unlearnability of SL-based attacks, we replace the data augmentation with contrastive augmentation and adjust the intensity of augmentation through a strength parameter, as shown in Appendix C.2, Code Listing 1.
> > > By default, we fix the augmentation strength as a constant value in the generation of AUE and AAP attacks, as discussed in Appendix D.4.
> > > Comprehensive experiments show that our attacks are both effective and efficient against SL and CL simultaneously.
> > >
> > >
> > >
> > > - **Annealing and tempering schemes**
> > > In the previous rebuttal, besides the constant augmentation scheme, we also try annealing and tempering schemes. These two dynamic choices are inspired by [1], which proposes that annealing down augmentation strength is beneficial for adversarial contrastive learning. We want to see if dynamic schemes also benefit our problem.
> > >
> > >     As shown in Figure 1 of our rebuttal, on CIFAR-10, the improvement of a particularly selected alternative scheme is marginal compared to the default one. Therefore, we believe that the **decisive factor** in our method is the contrastive augmentation itself, rather than the choice between constant and dynamic schemes for the augmentation.
> > >
> > > [1] Luo R, Wang Y, Wang Y. Rethinking the effect of data augmentation in adversarial contrastive learning. ICLR 2023.
> > >
> > >
> > > We hope our reply makes things more clear. Feel free to let us know if you still have concerns.

---

> > > > ### Comment · Reviewer_5WZD · 2024-08-10
> > > >
> > > > Thank you for your further explanations that addressed my concerns. I have adjusted my recommendation.

---

### Official Review · Reviewer_61jU · 2024-07-13

**Soundness:** 3
**Presentation:** 3
**Contribution:** 3
**Rating:** 5
**Confidence:** 3

**Summary:**

This paper aims to address the issue of effectively conducting availability attacks on both supervised learning (SL) and contrastive learning (CL) algorithms. Specifically, the paper highlights that existing methods fail to simultaneously achieve "unlearnability" for both SL and CL, posing risks to data protection. To tackle this challenge, the paper proposes a novel approach that employs contrastive-like data augmentation within the supervised learning framework to execute effective attacks on both SL and CL.

**Strengths:**

1. The proposed methods achieve better efficiency and Pareto improvement for both the SL and CL tasks.

2. the first work that demonstrates adding contrastive augmentation resilience with an SL-based surrogate can fool CL-based models.

**Weaknesses:**

1. The proposed AUE and AAP are all brittle under diffusion-based purification, which limits the practical usage of unlearnable examples in the real world.

2. Lack of discussion and comparison with recent work [1] that leverages CLIP latent as guidance for crafting transferable and label-agnostic perturbation, which could potentially achieve efficient perturbation crafting for both SL and CL tasks.

3. The results in Table 3 show that the proposed methods did not consistently outperform other methods under different SL and CL settings, which makes the stability of improvement questionable.

4. This work essentially works by crafting perturbation that resilience to augmentations for both supervised learning and contrastive learning, which is a conceptually simple extension of the transformation augmentation technique in REM [2]. One of the fundamental limitations of such a direction is that unauthorized trainer might leverage stronger transformations like super-resolution or diffusion-based augmentation [3] in their training pipeline.

Typos: Line 20: Change "particular, Huang et al. [20] reduces" to "particular, Huang et al. [20] reduced."

Ref.

[1]. One for All: A Universal Generator for Concept Unlearnability via Multi-Modal Alignment, ICML'24

[2]. Robust Unlearnable Examples: Protecting Data Against Adversarial Learning, ICLR'22'

[3]. DiffAug: Enhance Unsupervised Contrastive Learning with Domain-Knowledge-Free Diffusion-based Data Augmentation, ICML'24

**Questions:**

Nowadays, one of the most popular ways to build classifiers is to leverage CLIP as a feature encoder with or without additional head to do classification. Is the perturbation crafted from AUE and AAP transferable to this setting?

**Limitations:**

Yes

---

> ### Author Rebuttal · Authors · 2024-08-06
>
> We sincerely thank the reviewers for their thorough and constructive feedback! We aim to clarify and address your concerns through our detailed replies.
>
> **[Weakness1] Comparison with “One for All (14A)”** We believe that "14A" and our work are more orthogonal in nature and will explain it in detail.
>
> - The biggest difference between our method and “14A” is that we are studying different types of transferability. Specifically, “14A” studies **cross-dataset transferability**, meaning that the noise generator should produce UEs effective against supervised learning on different datasets. However, our paper investigates **cross-algorithm transferability**, meaning the generated UEs should be effective not only against SL but also against CL on the same dataset.
>
> - “14A” does not claim to work for CL, nor does it present any experimental results related to CL. Meanwhile, we don't claim that our attacks transfer well across datasets since they are based on dataset-specific optimization instead of a noise generator and directly applying off-the-shell perturbations to a different dataset is much more difficult and less realistic.
>
> **[W2] Non-consistent advantage** The reviewer mentioned that our proposed attacks do not consistently outperform baseline methods against different CL algorithms.
> We believe this phenomenon is **not a weakness** of our method; rather, it reflects the vulnerability of CL-based methods including CP, TUE, and TP, in terms of cross-algorithm transferability.
>
> - CL-based attacks show varying performance against different CL algorithms (see Table 11 in our paper). For example, since TP is generated using SimCLR, it achieves rather low accuracy against SimCLR evaluation, i.e., 6.7% for CIFAR-100, while our AUE achieves 13.6%. However, when facing a different evaluation algorithm, saying BYOL, TP’s accuracy drastically increases to 27%, which is higher than AUE’s 19.2%.
>
> - In contrast, our SL-based attacks demonstrate relatively stable performance against different CL algorithms and surpass CL-based baseline methods in worst-case unlearnability.
>
> **[W3] REM's augmentation** The reviewer mentioned that our technique is an extension of the technique used by REM. We believe there are some misunderstandings here and we will elaborate below.
>
> - “Expectation over transformation (EOT)” was proposed to make robust adversarial examples in the physical world[1], for example, a 3D-printed "adversarial" turtle that can be classified as a rifle from different views.
>
> - Then REM uses a modified EOT in which the transformation distribution only contains Crop and Horizontal Flip.
> Note that these transformations are common in standard supervised learning and not exclusive to adversarial training.
> The ablation study empirically shows EOT can improve the performance of REM against adversarial training.
>
> - We want to clarify that our method is **not an EOT variant** at all since it does not involve taking expectations.
> Moreover, our SL-based method contains contrastive augmentation that does not appear in SL.
> We leverage contrastive augmentation since it is a fundamental component of CL and our goal is to achieve unlearnability for CL.
>
> - In summary, in terms of both motivation and technical details, our method is not an extension of EOT, but a technique specifically designed to address a particular problem.
>
> [1] Athalye A, et al. Synthesizing robust adversarial examples. ICML 2018
>
> **[W5] Diffusion purification**
> - We acknowledge that at this stage of development, diffusion-based purification techniques have impacted all methods that aim to protect images through subtle perturbations, not only the method we studied, which uses availability attacks to prevent unauthorized data usage, but also methods that protect the copyrights of artists’ works [2,3,4].
>
> - In Appendix D.9 of our paper, we discussed these techniques. We consider the defensive capability of diffusion purification as a limitation of this work, and exploring how to overcome this limitation is a promising direction for future research, such as incorporating the diffusion process into perturbation generation.
>
> - Since the code in DiffAug's GitHub repository does not include implementations for commonly used datasets such as CIFAR and ImageNet, we needed to spend additional time modifying the code to adapt it to our problem.
> Once the results are available, we will update them in a subsequent version.
>
>
>
> [2] Shan S, et al. Glaze: Protecting artists from style mimicry by {Text-to-Image} models. USENIX Security 2023
>
> [3] Liang C, et al. Adversarial example does good: Preventing painting imitation from diffusion models via adversarial examples. ICML 2023
>
> [4] Hönig R, et al. Adversarial Perturbations Cannot Reliably Protect Artists From Generative AI.arXiv:2406.12027
>
> **[W6] Typo** Thank you for pointing out it. We will correct it in the next version.
>
> **[Question] CLIP** We conduct linear probing upon CLIP on CIFAR-10 following the official example in the GitHub repository. The CLIP encoder is pre-trained by OpenAI and fixed.
> - The following results show that both AUE and AAP can make CLIP-extracted representations of training data deviate from the true representation distribution. Compared to AUE, AAP achieves more unlearnability in such scenarios.
>
>     ||Clean|AUE|AAP|
>     |-|-|-|-|
>     |CIFAR-10|94.99|90.63|51.22|
>     |CIFAR-100|80.00|72.56|66.82|
>
> - The reason for this could be that, CLIP exhibits adversarial vulnerability, making it easy to find pixel-level perturbations that alter feature semantics [5]. Although our attacks were not designed specifically for CLIP, the perturbations they generate might share some commonalities with CLIP’s adversarial examples. Compared to AUE, AAP, as an adversarial example for SL, possibly plays a greater role in confusing the feature semantics extracted by CLIP.
>
> [5] Fort S. Adversarial examples for the OpenAI CLIP in its zero-shot classification regime and their semantic generalization

---

> > ### Comment · Reviewer_61jU · 2024-08-07
> > **Thank you for your detailed response**
> >
> > Thank you for your detailed response. Your response addresses most of my concerns. I will raise my score. But I am still looking forward to seeing the perturbation against DiffAug experiments, since, as you mention, it's a fundamental challenge in this field.

---

> > > ### Author Response · Authors · 2024-08-13
> > > **Discussion about DiffAug**
> > >
> > > Thank you for acknowledging our work.
> > > We will address your concern about DiffAug through additional experimental results.
> > >
> > > Since DiffAug's public repository only contains code for biology datasets, we sent emails to the authors requesting official implementations for vision tasks. Unfortunately, we have not received a response so far.
> > > Consequently, based on the paper and the existing code, we did our best to reproduce DiffAug on CIFAR-10. We used a ResNet-18 as the encoder backbone and a UNet as the diffusion backbone.
> > >
> > > We train the DiffAug on clean/poisoned training data and then perform linear probing upon the encoder. The following table shows the test accuracy. **Our attacks successfully transfer to DiffAug**. Specifically, compared to the non-attack case, AUE and AAP attacks reduce the test accuracy by 46.44% and 50.48% respectively.
> > >
> > > | Clean | AUE | AAP |
> > > |-------|-----|-----|
> > > | 81.88 |35.44|31.40|
> > >
> > > In summary, although DiffAug incorporates diffusion-based data augmentation in contrastive learning, our proposed attacks are still effective against it.

---

> > > > ### Comment · Reviewer_61jU · 2024-08-13
> > > >
> > > > Thank you for your detailed response and the effort in testing against DiffAug. Did you test the attack performance against more typical diffusion purification approaches like those SDEdit-based or super-resolution-based [1,2,3]?
> > > >
> > > > [1]. Diffusion Models for Adversarial Purification, ICML'22
> > > > [2]. Black-box Backdoor Defense via Zero-shot Image Purification, NeurIPS'23
> > > > [3]. Pixel is a Barrier: Diffusion Models Are More Adversarially Robust Than We Think

---

> > > > > ### Author Response · Authors · 2024-08-13
> > > > >
> > > > > Thank you for sharing the recent papers. We have not tested them. In the future version, we will discuss them in more detail in the relevant sections.

---

### Author Rebuttal · Authors · 2024-08-07

We would like to express our gratitude to the reviewers for their careful consideration and valuable feedback!

In our responses to each reviewer, we have clarified and addressed the weaknesses and issues raised, including many additional supplementary experiments.
Due to the word limit of the rebuttal and the design of the review system, we were unable to fully present the information from the tables and figures in our responses.

Here, we have consolidated the charts and tables from the supplementary experiments into an additional document and uploaded it.

- **Figure 1** Based on Reviewer 5WZD’s suggestion, we conducted more ablation studies on augmentation strategies. Figure 1 shows the experimental results of using a dynamic augmentation scheme for AUE.

- **Figure 2** Based on Reviewer 2’s suggestion, we conducted experiments with MAE. Figure 2 shows the training process of MAE fine-tuning under attacks.

- **Figure 3** The results of applying perturbations after augmentation for AUE, which is also an ablation study on augmentation strategies.

- **Table 1** To address Reviewer 3’s questions, we tested the effectiveness of our attacks against CLIP and showed results in Table 1.

- **Table 2** The performance of our attacks against MAE.

---

### Decision · Program_Chairs · 2024-09-25

**Decision:**

Accept (poster)

**Comment:**

The paper introduces two novel attack methods, AUE (Augmented Unlearnable Examples) and AAP (Augmented Adversarial Poisoning), designed to thwart both supervised learning (SL) and contrastive learning (CL) algorithms. It paper addresses a crucial gap in the field: while availability attacks have been proposed to prevent unauthorized use of private data in SL settings, they many fail to be effective against CL algorithms. The paper received a generally positive response from the reviewers, who appreciated its timely and relevant contributions. However, concerns were raised regarding the novelty and depth of the technical contributions, as well as the robustness of the proposed methods under certain conditions. The authors provided a comprehensive rebuttal addressing most of the concerns raised by the reviewers. They clarified the differences between their work and prior approaches, particularly emphasizing the distinctiveness of their focus on cross-algorithm transferability rather than cross-dataset transferability. These efforts were generally well-received by the reviewers, leading to score adjustments in favor of the paper. Overall, this paper makes a meaningful contribution to the field of availability attacks by proposing methods that are both effective and efficient across supervised and contrastive learning paradigms. It would benefit from a more modest presentation of its novelty and an expanded discussion on the theoretical implications and potential limitations of the proposed methods.